

# Effect of small-scale snow surface roughness on snow albedo and reflectance

Terhikki Manninen[1], Kati Anttila[1], Emmihenna Jääskeläinen[1], Aku Riihelä[1], Jouni Peltoniemi[2], Petri Räisänen[1], Panu Lahtinen[1], Niilo Siljamo[1], Laura Thölix[1], Outi Meinander[1], Anna Kontu[1], Hanne Suokanerva[1], Roberta Pirazzini[1], Juha Suomalainen[2], Teemu Hakala[2], Sanna Kaasalainen[2], Harri Kaartinen[2, 3], Antero Kukko[2, 4], Olivier Hautecoeur[5] and Jean-Louis Roujean[6]

[1]Finnish Meteorological Institute, Helsinki, P.O. Box 503, FI-00101, Finland
[2]Finnish Geospatial Research Institute/National Land Survey, Geodeetinrinne 2, 02430 Masala, Finland
[3] University of Turku, Department of Geography and Geology, 20500 Turku, Finland
[4]Aalto University, Department of Built Environment, 02150 Espoo, Finland
[5]Météo-France, currently at Exostaff GmbH/EUMETSAT
[6] Centre d'Etudes Spatiales de la BIOsphère (CESBIO) - UMR 5126 - 31401 Toulouse, France

*Correspondence to*: Terhikki Manninen (Terhikki.Manninen@fmi.fi)

**Abstract.** The primary goal of this paper is to present a model of snow surface albedo accounting for small-scale surface roughness effects. The model is based on photon recollision probability and it can be combined with existing bulk volume albedo models, such as TARTES. The model is fed with in situ measurements of surface roughness from plate profile and laser scanner data, and it is evaluated by comparing the computed albedos with observations. It provides closer results to empirical values than volume scattering based albedo simulations alone. The impact of surface roughness on albedo increases with the progress of the melting season and is larger for larger solar zenith angles. In absolute terms, surface roughness can decrease the total albedo by up to about 0.1. As regards the bidirectional reflectance factor (BRF), it is found that surface roughness increases backward scattering especially for large solar zenith angle values.

## 1    Introduction

The global energy budget is affected by surface albedo, which describes the level of brightness of the surface. Due to its central role for climate, it has been defined as an essential climate variable (ECV) by the Implementation Plan for Global Observing System for Climate in support of the United Nations Framework Convention on Climate Change (GCOS Secretariat 2006). The large areal coverage of seasonal snow, together with the high reflectivity of snow, contributes to the relevance of snow albedo on the global energy budget (Flanner et al., 2011; Mialon et al., 2005). Snow component is also important for the liveability of dry and cold areas for both humans and ecosystems by providing a source of melt water in spring and shelter and insulation in winter. Changes in the duration of snow cover and snow type are vital for people and ecology of these areas. Accurate large-scale monitoring of snow properties over large areas is only feasible in practice using satellite data-based methods. Prior to that, it is required to obtain a detailed understanding of the reflectivity and scattering properties of snow.



The surface reflectivity of snow depends on grain size, shape and impurity content, which are the basic properties for handling the volume scattering of snow. Traditionally snow grain size is characterized by the diameter, whereas it has been demonstrated

that the specific surface area (SSA) is the more appropriate variable to describe the scattering area per volume (Domine et al., 2012; Leppänen et al., 2015). Light attenuation within the snowpack is related to the density of the scattering elements per unit volume. In addition, layer structure, grain shape, anthropogenic and natural impurities (such as black carbon, dust and algae) and close-packing effects of snow grains affect scattering properties and thus, the albedo of a snowpack (Warren and Wiscombe, 1980; Kokhanovsky and Zege, 2004; Aoki et al., 2011; Kokhanovsky, 2013; Libois et al., 2013; Libois et al., 2014;

Komuro and Suzuki, 2015; Peltoniemi et al., 2015; Pirazzini et al., 2015; Räisänen et al., 2015; Cook et al., 2017; He et al., 2017, Kokhanovsky et al., 2018). Several models for the coupled mass and energy balances of snow on the ground have also been developed (Flanner and Zender, 2006; Essery, 2015). The decrease of snow albedo due to shadowing effects of larger-scale topography (Picard et al., 2020) and surface features such as sastrugi and crevasses have also been investigated both from the point of view of measurement and modelling (Leroux and Fily, 1998; Warren et al., 1998; Zhuravleva and Kokhanovsky,

2011; Lhermitte et al., 2014). But smaller-scale (mm to 10 cm) surface roughness has so far received poor attention in snow albedo modelling.

Snow grain size can also be related to micro-scale surface roughness. Initially snow surface is formed by falling snowflakes, which attach to the surface at first contact instead of being arranged according to the positions of minimum energy (Löwe et

al. 2007). Surface crystals are rearranged and shaped by the winds near the surface through saltation, which is the transport of snow in periodic contact with and directly above the snow surface. This process is governed by both the atmospheric shear forces and the moving snow particles (Pomeroy and Gray, 1990). The wind both breaks the particles into smaller pieces and helps the grains grow mass from the air moisture (Armstrong & Brun 2008). These atmosphere-surface interactions create some links between local small-scale surface roughness and the grain size properties of the topmost layers in the snowpack.

Moreover, the physical processes governing the snow grain metamorphism (temperature gradient, absorption of solar radiation, water vapor diffusion, liquid water formation) also affect the stickiness and, thus, the aggregation of grains (Löwe et al., 2007), which is associated to the formation of mm-cm-scale surface roughness.

If the surface were completely isotropic, the surface albedo might in many cases be well explained using only the grain size as

a descriptor of the snow pack of sufficient thickness to be semi-infinite from the scattering point of view. But typically, the surface structure slopes and snow properties influenced by wind are not identical in the windward and leeward sides (Sommer et al., 2018). This means that in clear-sky conditions the albedo will not necessarily be the same for azimuthally opposite viewing directions, when the saltation effect is marked. In addition, hoar frost formation depends more on the air temperature and humidity than the grain size of the existing snowpack. All in all, despite the dominant character of the snow grain size to



the scattering from a snowpack, the small-scale surface roughness has also a role independent of the snow grain size that should
be paid attention to. This study focuses on  the effect of surface roughness on snow albedo.

Here, a method taking into account the small-scale surface roughness in addition to the normal bulk volume scattering is
developed for the black-sky (directional–hemispherical reflectance DHR), white-sky (bihemispherical reflectance BHR in
isotropic diffuse illumination) and blue-sky albedo (bihemispherical reflectance BHR in ambient illumination) (Lucht et al.,
2000; Schaepman-Strub et al., 2006). The main points of the model are described in Section 3.2 and detailed equations are
derived in Appendix A. The TARTES snow model is used to simulate the albedo of a smooth snowpack (Warren, 1984; Warren
and Brandt, 2008; Kokhanovsky and Zege, 2004; Baldridge et al., 2009; Libois et al., 2013; Libois et al., 2016; Picard et al.,
2016).


The rough snowpack albedo model is tested with measurements carried out during the SNOw Reflectance Transition
EXperiment (SNORTEX) campaign (Roujean et al., 2010; Manninen and Roujean, 2014) in Sodankylä, Finnish Lapland in
March – April, 2009 and in March 2010 augmented with operational albedo measurements that Finnish Meteorological
Institute (FMI) carries out in The Arctic Space Centre of FMI in Tähtelä, Sodankylä. The physical properties of snow during
the campaign were measured from snow pit profiles. The modelled albedo is compared with measured albedo values in diffuse
and clear sky cases. The diverse snow measurements are briefly described in Section 2 and more details are available in the
given references. The high-resolution surface roughness profiles obtained using a scaled plate (Section 2.1) were also analysed
with ray tracing calculations to obtain the directional scattering characteristics related to the small scale surface roughness.
The BRF thus obtained was compared to empirical BRFs provided by FIGIFIGO measurements (Peltoniemi et al., 2005, 2015,
2014; Section 2.8). The varying role of the small-scale roughness from midwinter conditions throughout the melting season is
demonstrated in Section 4.

## 2    Data

### 2.1    Test area

Diverse properties of snow were measured in Sodankylä, northern Finland in March and April, 2009 and in March 2010 in an
area of about 10 km x 10 km (Figure 1, Manninen and Roujean, 2014). Every day about half a dozen test sites were located in
one land cover type (either forest or open areas, the latter being typically aapa mire). The last (first) measurement of the day
in 2009 (2010) was carried out in the NorSEN mast area (67.3621°N, 26.63445°E), which is located in similar terrain about
550 m from the place, where FMI conducts operational surface albedo measurements (67.36664°N, 26.628253°E downward,
95    67.36695°N, 26.62973°E reflected). Hence, the operational albedo values should be representative for the time series of the
snow pit measurements at the NorSEN mast.



## 2.2    Grain size and density profiles of snowpack

Measurements of snow depth, total density, water equivalent (SWE), humidity profile, temperature profile, grain size profile, surface roughness and surface impurity content were carried out at snow pits located in Sodankylä in an area with corner co-ordinates (67.36°N, 26.63°E; 67.45°N, 26.86°E) in March and April, 2009 and in March 2010 (Manninen and Roujean, 2014). In addition crystal size photos of the snow layers, surface roughness photos and photos of the top surface impurities were taken. In this study we concentrate on the values measured in 2009. The air temperature in March was mostly below 0°C, whereas in April it was almost all the time above 0°C (Table 1). Hence, April represents the melting season and March is still midwinter. This is also clear from the increase in median density of the snowpack and the decrease of median snow water equivalent value from March to April. About 40 snow pit measurement points were located in a larger area (67.42°N, 26.04°E; 67.85°N, 26.91°E), where the maximum measured snow depth was 92 cm in March and 76 cm in April. The total density varied in the range 0.18 … 0.32 g/cm³ in March and in the range 0.27… 0.57 g/cm³ in April. The corresponding variation ranges for the snow water equivalent were 20 … 250 mm and 34 … 239 mm, but in April there was plain water in several places in the snowpack. Hence, the area covered by the snow pit measurements represents the local variation of snow properties to a large extent.

The traditional snow grain size (the largest dimension of the snow grains, Fierz et al, 2009) was visually estimated using graded plates, collecting the snow crystals from 10 cm-thick snow layers from the bottom to the top of the snowpack. For each analysed sample of snow crystals, in addition to the typical value of the largest grain dimension also its minimum and maximum value were provided. The measured snow grain sizes differ from the optically equivalent snow grain size (Mätzler, 1997; Neshyba et al., 2003). To partly compensate for this, the minima of the largest grain diameter were applied in the radiative transfer calculations as the effective diameter. Although this causes some uncertainty in the interpretation of the computed absolute albedo values (particularly for the cases of fresh snow) it has much less impact on the derived effect of small-scale surface roughness on snow albedo. The density profile of the snowpack was measured for the same layer structure using the snow fork (Toikka, 1992). The variation range of the grain size and density is shown for the surface layer and for the whole snowpack in Table 2.

## 2.3    Surface roughness from plate measurements

The surface roughness up to 1 m scale was measured in March and April, 2009 and in March 2010 by taking photos of a graded plate placed perpendicularly in the snowpack (Figure 2). The snow surface profiles were automatically calculated from the photos using an image processing technique and the scale at the edge of the plate (Manninen et al., 2012). Control points at the scales were used both for the removal of the barrel distortion of the camera optics and transformation of the pixel coordinates to millimetres with photogrammetric methods. The plate surface roughness measurements were carried out at the





same sites as the snow pit measurements (Section 2.2). At each site, profiles were measured in two perpendicular directions
with a 1 m interval along 50 m to 100 m distance.

The surface profiles were used to derive the rms height and correlation and their distance dependence (Manninen, 2003; Anttila
et al., 2014). The snow surface roughness is close to a Brownian fractal surface (Anttila et al., 2014) so that the logarithm of
the rms height $\sigma$ depends linearly on the logarithm of the distance $x$ and the correlation length $L$ is linearly related to the
distance

$$\log \sigma = a + b \log x \tag{1}$$
$$L = k_0 + kx \tag{2}$$

where $a$, $b$, $k_0$ and $k$ are constants and $k_0 = 0$ for an ideal Brownian surface. For each profile the values of the constants were
calculated by linear regression using varying sliding window sizes (Anttila et al., 2014).

In addition, the rms slopes $\beta$ were calculated for the measured spatial resolution, which was on the average 0.26 mm. The
vertical resolution was about 0.1 mm and the horizontal resolution 0.04 mm (Manninen et al., 2012).

## 2.4    Surface roughness from laser scanning

In addition to the plate measurements, laser scanning data for snow roughness was utilized. The laser scanning data used in
this study has been acquired using the FGI ROAMER system (Kukko et al. 2007). The system, including a FARO Photon 120
laser scanner, a NovAtel SPAN GPS-IMU system, and data synchronizing and recording devices, was mounted on a sledge,
which was towed by a snow mobile. The data acquisition covers a 2.5 kilometres zone at each side of an official snow mobile
track (see Kukko et al. 2013 for more details). The landscape covered sparse pine forests and open bogs. The absolute precision
of these measurements was analysed by Kaasalainen et al. (2011) to be better than 5 cm, while the relative accuracy required
to observe the changes in the snow roughness were found to be 0.7 mm-2 mm for a static system, and less than 10 mm when
the snowmobile was moving. The best repeatability was achieved at ranges closer to the scanning system, i.e., below 5 m. The
data quality and precision were controlled using control points measured with a VRS-GPS (Leica SR530 receiver + AT502
antenna) precision GPS.

The laser profiles (about 16 profiles per 1m at 3 m/s snow mobile velocity) measured on March 18, 2010 were used to analyse
the variation of the slope angles in a larger area than was possible using the plate profiles. The dependence of slope angles on
distance between successive points was studied by calculating the root mean square (rms) slopes per size of horizontal distance





increment within the about 2.4 km long and 2 x 3.2 m wide scanned area. The scanned profiles on both sides of the snowmobile route were used.

## 2.5 Snow impurity content

The snow impurity was measured by filtering a melted sample of snow. The Quartz filters were analysed using NIOSH 5040 protocol. The increase of the median amount of impurities from March to April is obvious from Table 1. (Meinander et al., 2013; Meinander et al., 2014; Meinander et al. 2020).

## 2.6 Surface albedo

The surface albedo was operationally measured at Sodankylä (67.36664°N, 26.628253°E downward, 67.36695°N, 26.62973°E reflected) with a one minute interval using Kipp & Zonen CM11 Pyranometers. The site is surrounded by trees and houses, so that shadowing takes place in certain azimuth directions, when the solar elevation is very low. Hence, the measured white-sky albedo values are considered more reliable than the blue-sky values. The least shadowed azimuth direction in early March corresponded to the solar zenith angle value of 73° in the afternoon. Thus, the blue-sky albedo values used in the analysis were

all taken from the afternoon, when the solar zenith angle equalled 73°. This means that the azimuth direction used increased a bit during the spring, but it did not cause any additional shadowing problem. Yet, the clear-sky albedo of March 12 was replaced with the diffuse albedo dominating that day, because the clear-sky albedo value of a narrow time window seemed unrealistically small. Albedo values measured at the NorSEN mast on April 21, 2009 using a portable Kipp & Zonen CM 14 albedometer were used to calibrate the operationally measured albedo data in order to correct for the slight difference in

location of the upward and downward looking pyranometer used for operational measurements.

The portable albedometer was used in April 2009 to measure the snow surface albedo in the same areas where the snow pits were made (Section 2.2). The instrument was installed on a short boom affixed on a lightweight camera tripod for easy transport. The tripod legs affect somewhat the reflected radiation measurements, and therefore a first-order correction was

applied. It was based on estimation of the solid angle blocked by the tripod legs from fisheye lens photograph with a camera mounted onto the albedometer position on the tripod, assuming a constant albedo of 0.1 for the dark carbon fibre legs. The albedometer was calibrated against a reference pyranometer at FMI-Helsinki prior to each campaign. The albedometer was carefully levelled on the tripod before measurements at each location and the stability of levelling monitored regularly, as melting snow may become unstable during the day.






## 2.7  Spectral reflectance

The spectral reflectance of snow was measured using the ASD FieldSpec Pro JR spectrometer on several days, specifically in the perfectly overcast conditions on March 13 and on the perfectly clear-sky day April 22, during the campaign in 2009. The irradiance spectra were measured as well. Every spectrum is an average of 30 individual spectra. The spectrometer was
calibrated by the manufacturer prior to the campaigns. The instrument was powered on at least 15-20 minutes before each measurement to ensure an even operating temperature. A Spectralon panel was used as white reference for the reflectance measurements. Most measurements took place from a height of 50-60 cm with an 8-degree foreoptic. The spectrometer was optimized before each measurement.

## 2.8  BRF

The bidirectional reflectance factor BRF of snow was measured using the Finnish Geodetic Institute's Field Goniospectrometer FIGIFIGO (Peltoniemi et al., 2005, 2015, 2014).  FIGIFIGO consists of a motorized arm of length of 2 m, moving the optics head +-90 degree around nadir, and an ASD Field Spec PRO FR spectrometer recording the spectrum in the range of 350 – 2400 nm. The azimuth is turned manually, and all angles and coordinates are recorded automatically, based on inclination,
direction, and position sensors. The footprint is around 10 cm in diameter. FIGIFIGO gives spectrally resolved BRF data, relative to Spectralon reference standard, from which also spectral albedo can be evaluated, but as the system is not absolutely calibrated in the field setup, real broadband albedos and BRF need external solar spectrum. In the results shown, a mean solar spectrum is used that may differ several per cent from the real time one.

In Mantovaaranaapa, 3 sets of rough snow were measured, and one set of smoother snow formed by a thin layer of windblown grains. Another set of thin and rough snow was measured in Korppiaapa, but were not used in the present study. The sunlight measurements were complemented by set or artificial light measurements of smoother snow near the NorSEN mast.

## 3  Methods

## 3.1  Smooth snowpack albedo modelling

The TARTES model (available at: https://snowtartes.pythonanywhere.com/) was used to estimate the snowpack white-sky and black-sky albedo values (Warren, 1984; Warren and Brandt, 2008; Kokhanovsky and Zege, 2004; Baldridge et al., 2009; Libois et al., 2013; Libois et al., 2016; Picard et al., 2016). It is a fast and easy-to-use optical radiative transfer model and represents the snowpack as a stack of horizontal homogeneous layers. Each layer is characterized by the snow grain size, snow
density, impurities amount and type, and two parameters for the geometric grain shape: the asymmetry factor and the



absorption enhancement parameter. The albedo of the bottom interface can be prescribed (here 0.13), although the bottom interface only markedly impacts thin snowpacks (<5 cm depth). The model is based on the Kokhanovsky and Zege (2004) formalism. The required input values for the model (density and grain size profile) were provided by the snow pit measurements of the SNORTEX campaign (Manninen and Roujean, 2014; Section 2.2). The amount of impurities was

temporally interpolated from the values of measured days. The black-sky albedo values of the bulk snowpack were derived by weighting the spectral albedo with the standard top of atmosphere (TOA) spectrum ASTMG173. The white-sky albedo values of the bulk snowpack were derived by weighting the spectral albedo with measured diffuse irradiance spectra of the cloudy day March 13, 2009.

The black-sky albedo values were calculated for three local incidence angle values of each plate surface roughness profile: the mean, mean minus one standard deviation and the mean plus one standard deviation of the individual local incidence angle values determined for each slope of the surface roughness profiles. The nominal incidence angle was set to the solar zenith angle value at the time of the measurements of the plate surface roughness profiles and the density and grain size values of the snowpack layers. The blue-sky albedo values were obtained from the black-sky and white-sky albedo values using the fraction

of diffuse irradiance operationally measured at Tähtelä.

## 3.2    Rough snowpack albedo modelling

From the theoretical point of view there is a difference in scattering from a snowpack having an ideally planar surface and a rough surface, because the rough surface may have an incidence angle distribution that markedly differs from the Gaussian

distribution of incidence angles produced by a random volume of spherical scatterers partly shading each other. In addition, the roughness may cause a markedly higher amount of multiple scattering thus reducing the amount of radiation escaping the target.

Scattering from randomly rough continuous surfaces is related to the characteristic size of the surface roughness with respect

to the wavelength used (Beckmann and Spizzicchino, 1963; Ulaby et al, 1982; Tsang et al., 1985, Fung, 1994). When the surface roughness of a randomly rough continuous surface is large compared to the wavelength of the electromagnetic wave, the scattering of the wave from the surface can be approximated by scattering from random facets (i.e. using the Kirchhoff approximation), whose slopes determine the scattering directions. As the shortwave illumination covers the wavelength range of about 300 nm – 2500 nm, all structures in the mm scale (or above) are large compared to the wavelength, so that a facet-

based surface scattering calculation is reasonable. Each facet is taken to represent a volume of random scatterers and the local incidence angle of the incoming radiation is the angle between the normal of the facet and the solar zenith angle (Figure 3). The surface of a snowpack is not a continuous solid surface, but when the snowpack surface is rough, the incidence angle distributions of the scattering elements may deviate from that of a planar surface with randomly oriented scatterers. In addition,



it is possible that a photon escaping one facet hits another facet. The snowpack scattering can then be thought to have elements
both of bulk volume scattering and surface scattering. The following 2D analysis demonstrates this idea.

Multiple scattering between facets can be taken into account using the photon recollision probability theory (Knyazikhin et al.,
1998; Panferov et al., 2001; Smolander and Stenberg, 2005; Rautiainen and Stenberg, 2005; Stenberg et al., 2008; Stenberg
and Manninen, 2015; Stenberg et al., 2016). The formulation is shown in Appendix A separately for diffuse and direct
irradiance. The essential equations are repeated here. Firstly, for the diffuse case, the white-sky albedo $\alpha_w$ (Lucht et al., 2000,
Schaepman-Strub et al., 2006) is related to the average number of facet-to-facet scattering events $n$

$$\alpha_w = \alpha_{w0}{}^{n+1} \tag{3}$$

where $\alpha_{w0}$ is the white-sky albedo of the bulk volume.

Second, for direct illumination, the black-sky albedo $\alpha_b(\theta_i)$ (Lucht et al., 2000, Schaepman-Strub et al., 2006) is approximately
related to the $\alpha_{w0}$, $n$ and the average number $m(\theta_i)$ of facet-to-facet scattering events in direct illumination conditions by

$$\alpha_b = \alpha_{b0}\alpha_{w0}{}^n \frac{(1-\alpha_{w0}{}^{m+1})}{(1-\alpha_{w0}{}^{n+1})} \tag{4}$$

where $\alpha_{b0}(\theta_i)$ is the black-sky albedo of the bulk part of the snowpack (i.e. the albedo without the surface roughness
contribution). In this study the bulk albedo values $\alpha_{w0}$ are produced using the TARTES model (Section 3.1).

The albedo $\alpha$ in mixed illumination conditions is typically estimated using the weighted mean approximation of the two
extreme values $\alpha_w$ and $\alpha_b$ (Lucht et al., 2000, Schaepman-Strub et al., 2006; Román et al., 2010)

$$\alpha = f\alpha_w + (1-f)\alpha_b \quad , \tag{5}$$

where $f$ is the fraction of diffuse irradiance. According to Eqs. 3 - 5 the blue-sky albedo is then estimated from

$$\alpha = \left(f\alpha_{w0}{}^{n+1} + (1-f)\alpha_{b0}\alpha_{w0}{}^n \frac{(1-\alpha_{w0}{}^{m+1})}{(1-\alpha_{w0}{}^{n+1})}\right) = \alpha_{w0}{}^n\left(f\alpha_{w0} + (1-f)\alpha_{b0}\frac{(1-\alpha_{w0}{}^{m+1})}{(1-\alpha_{w0}{}^{n+1})}\right) \quad . \tag{6}$$

Obviously, surface roughness decreases the white-sky albedo and typically also the black-sky and blue-sky albedo. Only when
$m > n$, surface roughness can increase the black-sky albedo of bright targets. The effect of surface roughness is non-negligible





even when the roughness is not large. On the other hand, the larger the roughness is (i.e. the larger $n$ and $m$ are), the larger the effect is for darker targets. Hence, for snow the effect is larger in the near-infrared than in the visible wavelengths and in midwinter roughness alters the broadband albedo only slightly, whereas during the melting season, when the snow is darker, the effect of the roughness may be much larger. Thus, the effect of surface roughness would be larger for old snow than for
new snow, which may explain part of the quick darkening of snow during the melting season.

### 3.3    Ray tracing analysis of surface roughness

Scattering from the snow profiles measured using the plate was analysed by a ray tracing method using 1000 equally spaced rays per profile per direction. The number of hits $n_s$ on the surface (unity for single reflection and larger for multiple surface
scattering) and the direction of the escaping reflected ray was calculated as a function of the zenith angle of the incoming ray with an interval of two degrees from 0° to 80°. Scattering from the surfaces was calculated assuming completely smooth facets of continuous material, so that scattering from a facet was determined as mirror reflection. For each angle the case was calculated separately for rays coming from the left and from the right and the two results were unified to improve statistics. This choice was motivated by the known fact that, even when the snow is highly forward scattering, it is with respect to the
direction of the incoming solar radiation, not the wind direction. Thus, the dominant scattering direction moves with the sun during the day. In some cases, the ray was trapped to infinite reflection from facet to facet. But these quite rare ($< 1\,\%$) cases were excluded from the statistical analysis.

Late in spring the scattering from a snowpack often also contains a component that is something between volume and surface
scattering, namely deep narrow pits generated by impurities that have sunk downwards in the snowpack due to melting caused by absorption of solar radiation. This kind of an effect is not easy to take into account either in volume scattering or surface scattering, because they don't affect the roughness or density in a random way. Their contribution to albedo is beyond the scope of this study.

## 4    Results

### 4.1    Surface roughness and inputs for albedo modelling

To start with, we consider the snow surface roughness from the plate measurements. The rms slope angle calculated from the plate roughness measurements had an increasing trend from March to April (Figure 4). The surface height distributions developed towards a more Gaussian distribution from March ($R^2 = 0.97$) to April ($R^2 = 0.99$). This was evident also from the
change of the mean skewness value that was 0.4 in March and -0.12 in April.





The mean number of individual reflections $n_s$ per ray on the surface has an increasing trend during the melting season (Figure 5). It correlates well ($R^2 = 0.83$) with the rms slope angle ($\beta$) of the profile. A good general fit to all measured plate profile data was


$$n_s = 1 + 0.355332 \cos \theta_i{}^4 \beta - 1.08275\left(1 - \exp\left(1.75 \cos \theta_i{}^{1/4} \beta^4\right)\right) \qquad . \qquad (7)$$

The mean zenith angle to which the radiation escapes from the surface (when mirror reflection from the facet is assumed) is even more strongly correlated ($R^2 = 0.93$) with $\beta$ (Figure 6)


$$\theta_o = -0.925239 \left(1 - \frac{1}{(1+0.25\,\theta_i)^{\frac{3}{2}}}\right) - 1.29982 \left(1 - \frac{1}{(1+0.75\,\theta_i)^{\frac{3}{2}}}\right) \log(\beta) \qquad . \qquad (8)$$

All angles ($\beta$, $\theta_i$, $\theta_o$) are given in radians in the above equations, and $\theta_o > 0$ ($\theta_o < 0$) for forward-scattering (backward scattering). Obviously, the probability for backward scattering increases with increasing incidence angle and increasing $\beta$.

Since $\beta$ increases with time, also the probability for stronger backscattering from the snow cover increases with time. Indeed, it is well known that older snow cover is less strongly forward scattering than new midwinter snow (Peltoniemi et al., 2010).

Unfortunately, the $\beta$ values depend strongly on the scale of the measurements. The laser scanning data is well suited to demonstrate this, because the horizontal distance between successive data points increases from the beginning to the end of

the scan line. On March 18, 2010 plate profile measurements and laser scanning were carried out in the same relatively flat wetland area Mantovaaranaapa (67.4°N, 26.7°E). The laser scanning data covered a 2.4 km long and about 3.2 m wide area on each side of the snowmobile route. Altogether 10 plate profile measurements were taken directly after the scanning, at about 100-200 m intervals starting from the western edge of the scan route (Kukko et al. 2013). The average rms slope angle of the 10 plate profiles was 30.7° ($\beta = 0.54$) with an 80 % variation range of 24.7°… 34.3° ($\beta = 0.43$ … 0.60). Consequently, $n_s$

would then vary in the range 1…1.5 (Figure 5). The rms slopes were calculated also from the laser scanning profiles as a function of horizontal increments, which were within the range 5 mm … 100 mm. The number of points per distance varied in the range 4 thousand … 2.2 million. Nonlinear regression to the 36 points in the range 5 mm … 100 mm produced an exponential curve that approaches the mean rms slope value obtained from the 10 plate profiles of the same area and (Figure 7). Shorter increments of the laser data could not be reliably used in the analysis.


As the laser scanner covered a much larger area than the plate profiles, that data gives an estimate of the rms slope variation in a larger area and is based on a larger number of individual slope values. For the laser data set, the median difference between the 90 % quantile curve of the slope angle values and the rms slope angle value curve was 6.1°. The corresponding median





difference for the 10 % quantile curve from the rms slope angle curve was -5.9°. For the plate profiles, 90% and 10 % quantile

values of the rms slope angle differed from the mean rms slope value by 3.6° and -6.0°. Thus, the larger area covered by the

laser scanner shows a larger variation of the rms slope angle, as one could expect. Obviously, the laser scanner data can be

extrapolated to estimate $\beta$ at higher horizontal resolution than the measurements directly enable, but then $\beta$ has to be analysed

as a function of the horizontal distance increment (Figure 7). However, the strong variation of $\beta$ with spatial resolution suggests

that using less scale-dependent surface roughness descriptors would be desirable, if they just can provide the information

needed.

The relationship between other surface roughness parameters (such as rms height $\sigma$ and correlation length $L$) and $\beta$ is in general

not strong even for a Gaussian surface height distribution (Beckmann & Spizzicchino, 1963). For the whole period (March 3

– April 28, 2009) the ratio of $\sigma/L$ (determined for 60 cm distance) correlated however relatively well with $\beta$, the $R^2$ values

being 0.70, 0.62 and 0.67 for the whole data range, March and April, respectively, but the best descriptor of $\beta$ was found to be

$\sigma/b$ (Eq. 1), its $R^2$ values for the linear correlation being 0.78, 0.68 and 0.82 for the whole data range, March and April,

respectively. $\beta$ tends to increase with the progress of the melting season (0.002 radians per day). Likewise, its correlation with

$\sigma/b$ increases during the melting season. It was therefore examined, whether the measured surface albedo correlates well with

the measured surface roughness parameters. Using just the rms height (derived for a 60 cm horizontal scale) as an explanatory

variable of the albedo the coefficient of determination was $R^2 = 0.81$. The relationship between the albedo and surface

roughness parameters that are scale-independent in a large range (Manninen, 2003) was then evaluated. Indeed, a simple linear

regression for the data of March and April, 2009 produced a coefficient of determination value as high as $R^2 = 0.90$, when the

parameters $b$ and $k_0$ (see Eqs. 1 and 2) were used as explanatory variables (Figure 8). This result supports the view that surface

roughness affects the albedo.


## 4.2 Snow albedo spectra: measured vs. modelled

Two examples of snow nadir reflectance spectra measured with the ASD spectroradiometer are shown in Figure 9. On March

13, the sky was completely overcast, whereas on April 22 it was perfectly clear. For comparison, corresponding albedo spectra

modelled using TARTES are shown. The ASD reflectance spectra were scaled so that the derived broadband reflectance value

matched the calibrated operationally measured broadband albedo value. No BRF was available for the clear-sky case for the

location of ASD, but for old rough snow in the area it was relatively flat (see Section 4.5), so that the comparison of the spectral

reflectance and albedo values seems reasonable enough to enable the choice of the grain shape to be used in the TARTES

model calculations. The spectra modelled with TARTES accounted for the empirical grain size and density values, as well as

for black carbon but not for organic carbon (Table 1), which included needles and various tree trash deposited on snow. In

March, the impurity content in surface snow was very low, while in April, it was roughly 3 times higher (Table 1). In March a better fit is obtained using fractal grains, which result is also supported by photos taken of snow grains and the fact that the snow was fresh. In April the modelled albedo favoured the use of spherical grains rather than fractals in the calculations. The photos taken about the snow grains also supported the use of spherical grains in April. Thus, the TARTES results seem to

represent the snowpack in March well, but in April when the melting has been going on for a longer time the modelled albedo values are higher than the empirical reflectance values both in the visible and NIR wavelengths (less than ~1 μm), which dominate the value of the broadband albedo of snow.

### 4.3 Broadband albedo: measured vs. modelled

The evolution of the operationally measured broadband albedo in Tähtelä is shown for the periods March 12-19 and April 21-28 in Figure 10 together with the corresponding albedo values simulated using the TARTES model and the grain size and density measurements of the day in the Sodankylä area. Following the justifications outlined above, fractal grain shapes were used in March and spheres in April. The simulated values tend to exceed the measured ones, especially in April. Besides, the variation range of simulated albedo values is rather small compared to that of the empirical broadband reflectance values.

However, it will be demonstrated next that taking into account the surface roughness decreases the difference between simulated and empirical albedo estimates.

The albedo model taking into account both volume and surface scattering (Eqs. 3 - 5) was applied so that $\alpha_{w0}$ was the value provided by the TARTES model based on the measured density and grain size profile and impurity content. The values for $n$

and $m$ were derived from the empirical values of $n_s$. Namely, $m = n_s$ -1 for the local solar zenith ($\theta_{il}$) angle range derived using the ray tracing method. And $n$ is the weighted mean of $n_s$ -1, where the weights are $\cos(\theta_{il}) \cdot \sin(\theta_{il})$. The results are shown in Figure 11 and Figure 12.

First, the ratio $\alpha_w/\alpha_{w0}$ of the total white-sky albedo and the bulk white-sky albedo provided by the TARTES model is considered

in Figure 11. In March, this ratio varies mainly between 0.97 and 0.99, indicating that small-scale surface roughness decreases the snow albedo typically by 1-3%. With the progress of snow melt, the effect of surface roughness increases markedly. On 26-27 April (Julian days 116-117), the median of $\alpha_w/\alpha_{w0}$ falls below 0.9, indicating an over 10% decrease in snow albedo. The relative difference between the total and bulk albedo values is about the same for the black-sky case as for the white-sky case, but the solar zenith angle naturally slightly complicates that relationship (see Eqs. 3 and 4).


The modelled albedo values are further compared to observations in Figure 12. The variation range of the simulations shown with the background shading is based on the variation of the grain size, density and local incidence angle of the measured





profiles. Overall, the inclusion of surface roughness improves significantly the agreement of modelled albedos with the observed ones. A notable overestimation remains, however, on Julian days 77-78, and will be discussed in Section 5. All in all, the model is robust enough to be reasonably applied to empirical data of grain size, density and surface roughness.

Obviously, taking into account the surface roughness contribution improves the match of empirical and modelled results, but still it is very clear that the grain size, grain shape and SWE (or density) are dominant parameters, because the amount of volume scattering also affects directly the amount of surface scattering. The variation range of the modelled albedo is much larger for clear-sky cases than diffuse cases, which is understandable as some of the variation in clear-sky cases comes from the solar zenith angle variation during the day.

### 4.4 Albedo during snow metamorphosis on April 22

The modelled albedo results were compared with the Kipp & Zonen CM14 albedometer measurements carried out in Mantovaaranaapa on April 22, 2009, which was a perfectly clear day (Figure 13). One snow pit was measured during 9 – 10 UTC at 67.40735°N, 26.72357°E. The albedometer was positioned in its vicinity and it recorded the metamorphosis process shown by the linear ($R^2 = 0.998$) decrease of the albedo with time. Surface roughness of 54 individual profiles were retrieved with the plate method with typically 10 m incremental distance in perpendicular directions covering an area of about 100 m x 100 m (Figure 2). Each position of the plate was photographed three times, so that 18 separate profiles were characterized. Hence, the modelled albedo results were averaged to get one value per surface sample. The light grey band in Figure 13 shows the variation range of the broadband-converted reflectance spectra measured with the ASD spectrometer in the same area at the same time. Obviously, the modelled profile albedos fit in that range. The mean of the empirical albedo values is 0.66 and the modelled mean values are 0.72 and 0.68 for volume scattering only and for both volume and surface scattering, respectively. Taking into account the snow surface roughness thus improved the average modelled albedo estimate.

### 4.5 BRF

Since the contribution of surface roughness to the total albedo is markedly smaller than the contribution of the bulk volume scattering, it is clear that the volume scattering dominates also the BRF. However, the contribution of surface roughness is not negligible and the BRF of the surface scattering component may differ markedly from the bulk volume component, resulting in a complex total BRF. The ray tracing based surface BRFs (without any volume scattering contribution) were compared with empirical BRFs measured using FIGIFIGO (Figure 14) in Mantovaaranaapa in April 22, 2009. The area is an aapa mire, which late in spring affects the snow properties markedly (Figure 15). Sporadically the snow had melted and refrozen. Since the surface roughness profiles produce only 2D information and scattering angles differ markedly in 2D and 3D, the calculated ray tracing based 2D BRFs were converted to 3D versions assuming that each facet has besides the measured vertical angle





445    also an azimuth angle obeying a random uniform distribution between 0 … 180°. In fact, calculations were made for the range
0 … 90°, assuming the case to be symmetrical with respect to azimuth angle, like in constructing the FIGIFIGO based BRFs.
The 3D conversions make the peaks of the 2D scattering angle distributions slightly less distinct. No atmospheric contribution
was included in either data set.

The comparison of the ray tracing and surface roughness based BRFs with the empirical FIGIFIGO-based BRFs were made
in the principal plane, since the azimuth information of the former ones was just a statistical assumption to convert the 2D
principal plane BRF to 3D principal plane BRF. The surface scattering BRFs are typically peaked to the direction $\theta$ of forward
scattering matching mirror reflection of the surface plane of the snowpack. In addition, a backward peak in the direction $\theta$ -
90° is also strong in most cases (Figure 16). The balance between forward and backscattered intensity varies with incidence
angle so that large incidence angles favour surface backscattering due to roughness (Figure 17, Table 3). Although mere surface
scattering would lead to dominantly backward scattering in Mantovaaranaapa due to the large incidence angle values (55° …
64° for FIGIFIGO BRFs), the empirical BRFs measured using FIGIFIGO dominated by the volume scattering are still
dominantly forward scattering. The balance between forward/backward volume scattering is related to the grain shape
(Peltoniemi et al., 2010). But indeed, the smoothest snow sample produced least backscattering, the ratio of the backward to
the forward scattered amount of radiation was 0.58, whereas the corresponding ratio was on the average 0.73 for the rough
BRFs. This result is quite in line with the general ray tracing analysis results that surface roughness increases the fraction
scattered backwards (Table 3). Also, in a previous theoretical study of Gaussian surfaces it has been shown that roughness
affects the maximum direction of backward scattering (Jämsä et al., 1993). For the profiles measured in Mantovaaranaapa on
April 22 the ratio of the backward and forward scattered radiation amounts in the incidence angle range of the FIGIFIGO
measurements varied from 1.0 to 1.46. One has to take into account that the plate profiles register roughness in 1 m scale,
whereas FIGIFIGO measures samples of 10 cm diameter. Hence the largest spatial roughness may not necessarily show up as
strongly in the FIGIFIGO results.

## 5    Discussion

In this study, the equations combining the volume scattering and surface scattering were derived using the photon recollision
theory (Appendix A), because this theory could be extended to include the surface scattering effect. However, to describe
properly the volume scattering of real snowpacks, it is essential to pay attention also to layer structure, grain shapes and various
types of impurities etc., so that a more complex description is typically needed for realistic volume scattering estimation. In
principle, the photon recollision theory could be extended to take the layer structure of the snowpack into account by just
letting $p$ to be a function of the depth, i.e. $p = p(z)$, where $z$ would be the distance from the bottom or top surface. However,
that is beyond the scope of this study. Hence, the surface scattering part is developed so that in principle it can be combined



with any volume scattering method. One just applies the estimates of $\alpha_{w0}$ and $\alpha_{b0}$ derived with the chosen volume scattering model in Eqs. 3 - 5. Hence, to obtain more realistic volume scattering estimates for the snowpack, we used the TARTES model for volume scattering in the simulations.


The ray tracing analysis showed that the backward scattering increases with increasing surface roughness and increasing incidence angle of the illumination. However, that analysis concentrates only on surface scattering without any volume scattering contribution. Combining the surface and volume scattering contributions to BRF, perhaps using an adding procedure for radiative transfer, would be an interesting topic for future work.


Although the surface scattering model used here includes multiple scattering from the surface, it may be that a surface layer containing very deep pits would benefit from some special attention. Namely, surface roughness measurements methods are usually designed for typical roughness of about the same variation range horizontally and vertically, not for extremely deep pits. To some extent the pit structure will be taken into account by the volume scattering models, since they affect the density

of the surface layer of the snowpack. However, their very anisotropic (vertical) orientation is not well described by random scattering of a layer with reduced density. The pits act like illumination traps so that a larger part of illumination reaches lower layers of the snowpack before it is absorbed or scattered upwards. Therefore, the bulk albedo is smaller than for a completely random volume of the same density.

An example case is offered to illustrate this effect. Slight snow precipitation took place on March 16 so that a very fluffy surface structure of large dendritic snow crystals was formed on the snowpack (Figure 18 and Figure 19). The rms slope based on the plate method showed an increase with time from about 0.38 to about 0.63 with $R^2 = 0.68$. The rms height and correlation length also manifested clear evolution during those days (Figure 20). A related change is obvious also using the roughness parameters $a$ and $b$ (Anttila et al., 2014). Yet, the surface roughness measurements based on the plate method or laser scanning

are not able to catch the deep pit structure of the surface, because of shadowing effects. Therefore, the simulated albedo is higher than the empirical one (Figure 12, Julian days 76-78). However, even if the 2D-surface roughness were characterized properly, the surface scattering model based on a statistical approach of random scatterers would not be ideal for a case with a distinct periodic surface structure. For example, one could analyse separately the percentage of illumination that will be completely trapped by the deep pits and reduce the simulated total albedo with that fraction.


The plate profile method has the advantage of high spatial resolution. Its main drawback is that it can be used only, when the snow is relatively soft. It has been successfully used in Finnish Lapland (Anttila et al., 2014) and at Greenland Summit (Manninen et al., 2016), but Antarctic snow is typically so hard that it is not possible to immerse the plate in it. In addition, icy and crusty snow surface of Finnish Lapland in 2018 and 2019 turned out to be too hard for the plate. For laser scanning,

however, the hardness of the snowpack does not cause any problems. Indeed, laser scanning shows great potential for





measuring snow surface roughness as it can cover large areas with high point precision accuracy. It is a particularly good method for measuring larger-scale roughness from 5-10 cm upwards. The limiting factor in finer scale roughness measurements is the data resolution and footprint size of the laser beam. So far, the scanners with the highest point density, accuracy and smallest spot size are meant for indoors use, but as the technology improves, smaller and smaller features become

measurable also outdoors. In addition, the fractal nature of snow surfaces enables extrapolation of surface roughness from cm scale to mm scale (Kukko et al., 2013). Another benefit of using laser scanning for surface roughness measurements is that it leaves the surface intact. This enables repeatable measurements of the same surface giving a means to study the evolution of surfaces in time. The backscattering intensity of the laser beam is typically stored for each point measured by laser, and in the most modern scanners also the range deviation is stored. These features have so far not been widely used, but it could

potentially be used in the future for surface scattering property measurements and snow surface classifications.

## 6    Conclusions

A method was developed to model the effect of surface roughness on albedo besides the volume scattering. It can be combined with any volume scattering model. Applying measured surface roughness values to the model produced results closer to

measured values than only volume scattering simulations made with the TARTES model. The surface roughness is described by the average number of surface scattering events per ray, which is currently estimated from the rms slope values of the measured surface roughness profiles. High empirical correlation ($R^2 = 0.9$) of albedo with just two surface roughness related parameters supports the importance of surface roughness to albedo.

The albedo modelling results taking into account also the surface roughness indicate that it may decrease the albedo by about 1-3 % in midwinter and even more than 10 % during late melting season. The effect is largest for low solar zenith angle values and lower bulk snow albedo values. Hence, the effect is larger early and late times of day everywhere and it increases during the melting season especially at high latitudes, where the sun elevation is lower. Increasing surface roughness also favours more backwards scattering.


0

0

0

0

0

0

0

0

0

0

0

0

0

0

0

0

0

0

0

0

0

0

0

0

0

0

0

0





## Appendix A: Deriving the formulas for surface roughness effect on scattering

### Scattering of diffuse radiation

540 The scattering of light in canopies has for several years successfully been described with spectral invariants and the so-called photon recollision theory, $p$-theory, (Knyazikhin et al., 1998; Panferov et al., 2001; Smolander and Stenberg, 2005; Rautiainen and Stenberg, 2005; Stenberg et al., 2008; Stenberg and Manninen, 2015; Stenberg et al., 2016). The central parameter, the photon recollision probability $p$ is spectrally invariant and depends on the amount of scattering surface in the volume. Canopies don't have distinct upper surfaces, hence the $p$-theory is developed so far only for a scattering volume, but it has already been 545 successfully combined with forest floor scattering also for snow covered cases (Manninen and Stenberg, 2009). Here the $p$-theory is applied to snowpack scattering taking into account that the snowpack has a distinct surface, which may be rough.

A simple way to take into account the surface roughness effect on scattering is to consider every facet of the snowpack as a separate volume of scatterers. When the irradiance $i_0 = i_0(\lambda)$ first arrives at the facet, it enters a volume scattering sequence, 550 which can be described with the spectrally invariant photon recollision probability $p$ of the bulk part of the snowpack and the single-scattering albedo of the snow grains $\omega = \omega(\lambda)$ (Knyazikhin et al., 1998; Panferov et al., 2001). The radiation absorbed and scattered by the volume of the facet $a_0$ and $s_0$, respectively, are (Smolander and Stenberg, 2005; Rautiainen and Stenberg, 2005; Stenberg and Manninen, 2015)

$$a_0 = \frac{1-\omega}{1-p\omega} i_0 \tag{A1}$$

555 $$s_0 = \frac{\omega-p\omega}{1-p\omega} i_0 \qquad . \tag{A2}$$

For simplicity the dependence of $\omega$, $i_0$, $a_0$ and $s_0$ on the wavelength $\lambda$ is not shown explicitly in the equations. The radiation escaping the volume of the facet either escapes altogether or hits another facet and experiences another volume scattering sequence. The probability of the latter case is defined to be the surface photon recollision probability $p_s$. Because the snow grains in the bulk part are completely surrounded by other snow grains while in the surface almost only half of the surrounding 560 volume may contain snow grains, it is essential to assume that $p$ and $p_s$ are not identical.

The radiation escaping the snowpack altogether without hitting another facet, $r_0$, is

$$r_0 = (1 - p_s)qs_0 = (1 - p_s)q\frac{\omega-p\omega}{1-p\omega} i_0 \qquad , \tag{A3}$$

where $q = q(\lambda)$ is the fraction of the volume scattering escaping upwards. Essentially it corresponds to $Q$ defined by Stenberg et al. (2016, Eq. 24), but as it does not contain the fraction scattered upwards by the surface, it is not the total upwards scattered 565 fraction of light. Hence $q$ is used here instead of $Q$. Theoretically the values for $q$ are in the range 0 … 1 being the larger the thicker and denser the scattering layer is. The radiation hitting another facet is

$$i_1 = p_sqs_0 = p_sq\frac{\omega-p\omega}{1-p\omega} i_0 \qquad . \tag{A4}$$



The absorbed ($a_1$) and scattered ($s_1$) amounts of radiation by the second volume scattering sequence are

$$a_1 = \frac{1-\omega}{1-p\omega} i_1 = p_s q \frac{(1-\omega)(\omega-p\omega)}{(1-p\omega)^2} i_0 \tag{A5}$$

$$s_1 = \frac{\omega-p\omega}{1-p\omega} i_1 = p_s q \left(\frac{\omega-p\omega}{1-p\omega}\right)^2 i_0 \tag{A6}$$

The amounts of radiation escaping ($r_1$) and entering the following scattering sequence ($i_2$) of another facet are

$$r_1 = (1-p_s)q s_1 = (1-p_s)p_s q^2 \left(\frac{\omega-p\omega}{1-p\omega}\right)^2 i_0 \tag{A7}$$

$$i_2 = p_s q s_1 = p_s{}^2 q^2 \left(\frac{\omega-p\omega}{1-p\omega}\right)^2 i_0 \tag{A8}$$

Formulas for the corresponding radiation components in the following facet-to-facet scattering round are

$$a_2 = \frac{1-\omega}{1-p\omega} i_2 = p_s{}^2 q^2 \frac{(1-\omega)(\omega-p\omega)^2}{(1-p\omega)^3} i_0 \tag{A9}$$

$$s_2 = \frac{\omega-p\omega}{1-p\omega} i_2 = p_s{}^2 q^2 \left(\frac{\omega-p\omega}{1-p\omega}\right)^3 i_0 \tag{A10}$$

$$r_2 = (1-p_s)q s_2 = (1-p_s)p_s{}^2 q^3 \left(\frac{\omega-p\omega}{1-p\omega}\right)^3 i_0 \tag{A11}$$

$$i_3 = p_s q s_2 = p_s{}^3 q^3 \left(\frac{\omega-p\omega}{1-p\omega}\right)^3 i_0 \tag{A12}$$

Further on, the components corresponding to the $j$th round are

$$a_j = \frac{1-\omega}{1-p\omega} i_j = p_s{}^j q^j \frac{(1-\omega)(\omega-p\omega)^j}{(1-p\omega)^{j+1}} i_0 \tag{A13}$$

$$s_j = \frac{\omega-p\omega}{1-p\omega} i_j = p_s{}^j q^j \left(\frac{\omega-p\omega}{1-p\omega}\right)^{j+1} i_0 \tag{A14}$$

$$r_j = (1-p_s)q s_j = (1-p_s)p_s{}^j q^{j+1} \left(\frac{\omega-p\omega}{1-p\omega}\right)^{j+1} i_0 \tag{A15}$$

$$i_{j+1} = p_s q s_j = p_s{}^{j+1} q^{j+1} \left(\frac{\omega-p\omega}{1-p\omega}\right)^{j+1} i_0 \tag{A16}$$

The total amounts absorbed and scattered by the surface and volume are then

$$a = \sum_{j=0}^n a_j = \sum_{j=0}^n p_s{}^j q^j \frac{(1-\omega)(\omega-p\omega)^j}{(1-p\omega)^{j+1}} i_0 = \frac{(1-\omega)}{(1-p\omega)} i_0 \sum_{j=0}^n \left(p_s q \frac{\omega-p\omega}{1-p\omega}\right)^j \tag{A17}$$

$$s = \sum_{j=0}^n s_j = \sum_{j=0}^n p_s{}^j q^j \left(\frac{\omega-p\omega}{1-p\omega}\right)^{j+1} i_0 = \left(\frac{\omega-p\omega}{1-p\omega}\right) i_0 \sum_{j=0}^n \left(p_s q \frac{\omega-p\omega}{1-p\omega}\right)^j. \tag{A18}$$

And the total upwards escaping radiation is

$$r = \sum_{j=0}^n r_j = \sum_{j=0}^n (1-p_s)p_s{}^j q^{j+1} \left(\frac{\omega-p\omega}{1-p\omega}\right)^{j+1} i_0 = (1-p_s)q \left(\frac{\omega-p\omega}{1-p\omega}\right) i_0 \sum_{j=0}^n \left(p_s q \frac{\omega-p\omega}{1-p\omega}\right)^j. \tag{A19}$$

When the surface of snow is very smooth, the number of additional facet scattering sequences ($n$) the photon has before it escapes altogether may not be very large. When $n = 0$ and $p_s = 0$, there is no facet-to-facet scattering, i.e. the case is the normal





volume scattering case. The total absorbed and upwards escaping radiation of the snowpack are derived as infinite geometrical sums and are

$$a = \frac{(1-\omega)}{(1-p\omega)} \frac{1}{1-\left(p_s q \frac{\omega-p\omega}{1-p\omega}\right)} i_0 \tag{A20}$$

$$r = (1 - p_s)q\left(\frac{\omega-p\omega}{1-p\omega}\right)\frac{1}{1-\left(p_s q \frac{\omega-p\omega}{1-p\omega}\right)} i_0 \tag{A21}$$

The total white-sky (diffuse) albedo $\alpha_w$ of the snowpack is then

$$\alpha_w = \frac{r}{i_0} = (1 - p_s)q\left(\frac{\omega-p\omega}{1-p\omega}\right)\frac{1}{1-\left(p_s q \frac{\omega-p\omega}{1-p\omega}\right)} \tag{A22}$$

The white-sky (diffuse) albedo of the bulk part of the snowpack (without facet-to-facet scattering) $\alpha_{w0}$ is

$$\alpha_{w0} = q\frac{s_0}{i_0} = q\frac{\omega-p\omega}{1-p\omega} \tag{A23}$$

Hence, the total white-sky albedo is simply

$$\alpha_w = (1 - p_s)\alpha_{w0}\frac{1}{1-p_s\alpha_{w0}} \tag{A24}$$

Estimation of the parameter $p_s$ is not trivial, but when the average value for $n$ is known from measurements, then a reasonable estimate can be obtained from the total scattered energy (Eq. A18), which can then also be calculated using a finite geometric sum of $n$ terms with recollision probability of unity. This must then equal the probabilistic infinite sum formulation of Eq.

(A18). Taking into account Eq. (A23) the following relation is obtained

$$\frac{\alpha_{w0}i_0}{q}\frac{1-\alpha_{w0}^{n+1}}{1-\alpha_{w0}} = \frac{\alpha_{w0}i_0}{q}\frac{1}{1-(p_s\alpha_{w0})} \tag{A25}$$

Now the value of $p_s$ can be solved and is

$$p_s = \frac{1-\alpha_{w0}^n}{1-\alpha_{w0}^{n+1}} \tag{A26}$$

The relationship between the total albedo and the bulk albedo is then

$$\alpha_w = (1 - p_s)\alpha_{w0}\frac{1}{1-p_s\alpha_{w0}} = \alpha_{w0}^{n+1} \tag{A27}$$

When the surface does not cause additional scattering, $n = 0$ and the total albedo equals the bulk albedo. The larger the $n$ is the smaller is $\alpha_w$.

For a single photon the number of facet-to-facet volume scattering rounds is naturally an integer number. For the ensemble of the photons $n$ can be estimated to be

$$n = \sum_{j=0}^{\infty} p_s\left(\frac{s_j}{i_0}\right) = p_s\left(\frac{\omega-p\omega}{1-p\omega}\right)\sum_{j=0}^{\infty}\left(p_s q\frac{\omega-p\omega}{1-p\omega}\right)^j = p_s\left(\frac{\omega-p\omega}{1-p\omega}\right)\frac{1}{1-p_s q\frac{\omega-p\omega}{1-p\omega}} = \frac{p_s\alpha_{w0}}{q(1-p_s\alpha_{w0})} \tag{A28}$$

Combining Eqs. A26 and A28 it is possible to estimate $q$ for values $n > 0$ and it is

$$q = \frac{p_s\alpha_{w0}}{n(1-p_s\alpha_{w0})} = \frac{\alpha_{w0}(1-\alpha_{w0}^n)}{n(1-\alpha_{w0})} \tag{A29}$$

For $n = 1$, $q$ equals the bulk white-sky albedo and for larger (smaller) values of $n$ it is slightly smaller (larger) for medium albedo values.



**Scattering of direct radiation**

For the direct component of solar illumination one has to take into account that the irradiance depends besides the wavelength also on the solar zenith angle $\theta_i$, i.e. $i_0 = i_0(\lambda, \theta_i)$. The photon recollision probability in the bulk snowpack will be denoted for the first scattering sequence $p_1 = p_1(\theta_{il})$ and for latter sequences $p$, assuming that the incidence angle dependence is lost after the first volume scattering sequence. Here the local incidence angle of the facet is denoted by $\theta_{il}$. Then, the absorbed radiation of the volume will be (Stenberg and Manninen, 2015)

$$a_0 = \frac{(1-\omega)(1-\omega(p-p_1))}{1-p\omega} i_0 \tag{A30}$$

and the radiation scattered by the volume will be

$$s_0 = \frac{\omega(1-p_1-\omega(p-p_1))}{1-p\omega} i_0 \quad . \tag{A31}$$

When $p_1 = p$ the above formulas reduce to Eqs. A1 and A2 as they should.

The first surface scattering sequence shall be treated separately, because the volume scattering depends on the incident solar zenith angle. The surface photon recollision probability is denoted by $p_{s1} = p_{s1}(\theta_{il})$ for the first facet-to-facet scattering sequence and by $p_{sr}$ for the latter facet-to-facet scattering sequences. One should notice that $p_{sr}$ is not necessarily equal to $p_s$ of the diffuse case, although they certainly approach each other asymptotically. The reason for not taking them to be identical immediately is the typically small number of surface scattering events per ray of relatively smooth midwinter snow surfaces. The radiation escaping the facet altogether after just volume scattering, $r_0$, is

$$r_0 = (1 - p_{s1})q_0 s_0 = (1 - p_{s1})q_0 \frac{\omega(1-p_1-\omega(p-p_1))}{1-p\omega} i_0, \tag{A32}$$

where $q_0 = q_0(\lambda, \theta_i)$ denotes the fraction of the scattered radiation escaping upwards from the snowpack during the first volume scattering sequence. The radiation hitting another facet is

$$i_1 = p_{s1}q_0 s_0 = p_{s1}q_0 \frac{\omega(1-p_1-\omega(p-p_1))}{1-p\omega} i_0 \quad . \tag{A33}$$

The further scattering sequencies are assumed to be independent of the solar zenith angle of the original incident radiation. The absorbed ($a_1$) and scattered ($s_1$) amounts of radiation by the second volume scattering sequence are then

$$a_1 = \frac{1-\omega}{1-p\omega} i_1 = p_{s1}q_0 \frac{\omega(1-\omega)(1-p_1-\omega(p-p_1))}{(1-p\omega)^2} i_0 \tag{A34}$$

$$s_1 = \frac{\omega-p\omega}{1-p\omega} i_1 = p_{s1}q_0 \frac{\omega(\omega-p\omega)(1-p_1-\omega(p-p_1))}{(1-p\omega)^2} i_0 \quad . \tag{A35}$$

The amounts of radiation escaping ($r_1$) and entering the following scattering sequence ($i_2$) are

$$r_1 = (1 - p_{sr})q s_1 = (1 - p_{sr})p_{s1}q q_0 \frac{\omega(\omega-p\omega)(1-p_1-\omega(p-p_1))}{(1-p\omega)^2} i_0 \tag{A36}$$

$$i_2 = p_{sr}q s_1 = p_{sr}p_{s1}q q_0 \frac{\omega(\omega-p\omega)(1-p_1-\omega(p-p_1))}{(1-p\omega)^2} i_0 \quad . \tag{A37}$$



Formulas for corresponding radiation components in the following round are

$$a_2 = \frac{1-\omega}{1-p\omega} i_2 = p_{sr}p_{s1}qq_0 \frac{\omega(1-\omega)(\omega-p\omega)(1-p_1-\omega(p-p_1))}{(1-p\omega)^3} i_0 \tag{A38}$$

$$s_2 = \frac{\omega-p\omega}{1-p\omega} i_2 = p_{sr}p_{s1}qq_0 \frac{\omega(\omega-p\omega)^2(1-p_1-\omega(p-p_1))}{(1-p\omega)^3} i_0 \tag{A39}$$

$$r_2 = (1-p_{sr})qs_2 = (1-p_{sr})p_{sr}p_{s1}q^2q_0 \frac{\omega(\omega-p\omega)^2(1-p_1-\omega(p-p_1))}{(1-p\omega)^3} i_0 \tag{A40}$$

$$i_3 = p_{sr}qs_2 = p_{sr}{}^2 p_{s1}q^2q_0 \frac{\omega(\omega-p\omega)^2(1-p_1-\omega(p-p_1))}{(1-p\omega)^3} i_0 \qquad . \tag{A41}$$

The components corresponding to the $j$th round are

$$a_j = \frac{1-\omega}{1-p\omega} i_j = p_{sr}{}^{j-1}p_{s1}q^{j-1}q_0 \frac{\omega(1-\omega)(\omega-p\omega)^{j-1}(1-p_1-\omega(p-p_1))}{(1-p\omega)^{j+1}} i_0 \tag{A42}$$

$$s_j = \frac{\omega-p\omega}{1-p\omega} i_j = p_{sr}{}^{j-1}p_{s1}q^{j-1}q_0 \frac{\omega(\omega-p\omega)^j(1-p_1-\omega(p-p_1))}{(1-p\omega)^{j+1}} i_0 \tag{A43}$$

$$r_j = (1-p_{sr})qs_j = (1-p_{sr})p_{sr}{}^{j-1}p_{s1}q^jq_0 \frac{\omega(\omega-p\omega)^j(1-p_1-\omega(p-p_1))}{(1-p\omega)^{j+1}} i_0 \tag{A44}$$

$$i_{j+1} = p_{sr}qs_j = p_{sr}{}^j p_{s1}q^jq_0 \frac{\omega(\omega-p\omega)^j(1-p_1-\omega(p-p_1))}{(1-p\omega)^{j+1}} i_0 \qquad . \tag{A45}$$

The total amounts absorbed and scattered by the surface and volume are then

$$a = \sum_{j=0}^m a_j = \frac{p_{s1}q_0}{p_{sr}q} \frac{(1-\omega)(1-p_1-\omega(p-p_1))}{(1-p\omega)(1-p)} i_0 \sum_{j=0}^m \left(p_{sr}q\frac{\omega-p\omega}{1-p\omega}\right)^j \tag{A46}$$

$$s = \sum_{j=0}^m s_j = \frac{p_{s1}q_0}{p_{sr}q} \frac{\omega(1-p_1-\omega(p-p_1))}{(1-p\omega)} i_0 \sum_{j=0}^m \left(p_{sr}q\frac{\omega-p\omega}{1-p\omega}\right)^j . \tag{A47}$$

And the total upwards escaping radiation is

$$r = \sum_{j=0}^m r_j = (1-p_{sr})\frac{p_{s1}q_0}{p_{sr}} \frac{\omega(1-p_1-\omega(p-p_1))}{(1-p\omega)} i_0 \sum_{j=0}^m \left(p_{sr}q\frac{\omega-p\omega}{1-p\omega}\right)^j . \tag{A48}$$

where $m$ is the number of facet-to-facet scattering events. The total radiation escaping the snowpack upwards is derived again as an infinite geometrical sum

$$r = (1-p_{sr})\frac{p_{s1}q_0}{p_{sr}} \frac{\omega(1-p_1-\omega(p-p_1))}{(1-p\omega)} \frac{1}{1-\left(p_{sr}q\frac{\omega-p\omega}{1-p\omega}\right)} i_0 \qquad . \tag{A49}$$

The total black-sky (direct) albedo $\alpha_b$ of the snowpack is then

$$\alpha_b = \frac{r}{i_0} = (1-p_{sr})\frac{p_{s1}q_0}{p_{sr}} \frac{\omega(1-p_1-\omega(p-p_1))}{(1-p\omega)} \frac{1}{1-\left(p_{sr}q\frac{\omega-p\omega}{1-p\omega}\right)} \qquad . \tag{A50}$$

The black-sky (directional) albedo of the bulk snowpack $\alpha_{b0}$ is

$$\alpha_{b0} = q_0 \frac{s_0}{i_0} = q_0 \frac{\omega(1-p_1-\omega(p-p_1))}{1-p\omega} \qquad . \tag{A51}$$

Taking into account also Eq. (A23) the relationship between the total black-sky albedo and the bulk snowpack black-sky and white-sky albedo is





$\quad \alpha_b = (1 - p_{sr}) \frac{p_{s1} \alpha_{b0}}{p_{sr}} \frac{1}{1 - p_{sr} \alpha_{w0}}$ (A52)

Estimating $p_{s1}$ or $p_{sr}$ (whether equal to $p_s$ or not) is not trivial, but like in the case of diffuse irradiance one can benefit from measured value of $m$ by requiring that the total scattered energy (Eq. A47) is the same, whether it is calculated from the probabilistic infinite sum or a deterministic sum with recollision probability of unity, i.e.

$\frac{p_{s1}}{p_{sr}} \frac{1}{1 - p_{sr} \alpha_{w0}} = \left( \frac{1 - \alpha_{w0}^{m+1}}{1 - \alpha_{w0}} \right).$ (A53)

Unfortunately, there are two variables, $p_{s1}$ and $p_{sr}$, to solve, but only one equation. Hence, the theory does not provide an exact solution for both $p_{s1}$ and $p_{sr}$, but only $p_{sr}$ remains explicitly in the equation of the black-sky albedo

$\alpha_b = (1 - p_{sr}) \alpha_{b0} \left( \frac{1 - \alpha_{w0}^{m+1}}{1 - \alpha_{w0}} \right)$ (A54)

When the assumption that $p_{sr} = p_s$ is valid, the following formula is obtained for the black-sky albedo using Eq. (A26)

$\alpha_b = \alpha_{b0} \alpha_{w0}^n \left( \frac{1 - \alpha_{w0}^{m+1}}{1 - \alpha_{w0}^{n+1}} \right)$ (A55)

When $m = n$, the black-sky albedo reduces to $\alpha_{b0} \cdot \alpha_{w0}^n$. Further on, when $\alpha_{b0} = \alpha_{w0}$, the total black-sky and white-sky albedo values are equal as well, if $m = n$. The above approximation of the black-sky albedo is reasonable only, when the assumption $p_{sr} = p_s$ is good. Hence, further studies are needed for proper estimation of $p_{sr}$ and $p_{s1}$.

The number of facet-to-facet scattering rounds $m$ is estimated to be

$m = \frac{p_{s1} s_0}{i_0} + \sum_{j=1}^{\infty} p_s \left( \frac{s_j}{i_0} \right) = \frac{p_{s1} \alpha_{b0}}{q_0} - \frac{p_s \alpha_{b0}}{q_0} + \sum_{j=0}^{\infty} p_s \left( \frac{s_j}{i_0} \right) = \frac{(p_{s1} - p_s) \alpha_{b0}}{q_0} + \frac{p_{s1} \alpha_{b0}}{q(1 - p_s \alpha_{w0})}$ . (A56)

When $p_{s1} \rightarrow p_s$, $q_0 \rightarrow q$ and $\alpha_{b0} \rightarrow \alpha_{w0}$, $m \rightarrow n$, as it should. It should be noticed that the number of facet-to-facet scattering rounds in direct illumination ($m$) and in diffuse illumination ($n$) are not necessarily equal, the difference being related to the difference of $\alpha_{b0}$ and $\alpha_{w0}$ and $q$ and $q_0$. The estimate for $q_0$ can now be derived from Eq. (A56).



**Data availability**

Some data of the SNORTEX campaign is directly listed in the report (Manninen and Roujean, 2014). The plate snow profile
data and the ASD data are available upon request from Finnish Meteorological Institute. The FIGIFIGO measurements and
the laser scanning data are available upon request from Finnish Geospatial Research Institute/National Land Survey.

**Author contribution**

Albedo measurements: TM, AR

Albedo modelling: TM, EJ, AR, JP, RP, J-LR, OH, PR

Plate surface roughness: KA, TM

Laser scanning surface roughness: SK, HK, AKu, KA, TM

FIGIFIGO BRF: JP, JS, TH

Snow density and grain size profiles: PL, NS, LT, AKo

Snow impurities: AKo, HS, OM

ASD snow reflectance spectra: AR, AKo, HS, TM

Ray tracing analysis: TM

Synthesis of analyses: TM, JP, PR, JLR

Writing of the manuscript: everybody.

**Acknowledgements**

The co-operation with the operational units of FMI in Helsinki and in Sodankylä and all SNORTEX campaign participants is
gratefully acknowledged. The authors are indebted to Mr. Tuure Karjalainen for taking the plate surface roughness related
photos in the field in 2009. Panu Lahtinen, Niilo Siljamo and Laura Thölix took the other photos. The authors are also grateful
to Mr. Markku Ahponen and Mr. Veikko Mylläri for the snow depth, density and SWE measurements of the remote points
outside the intensive test area and to Mr. Markku Ahponen, Mr. Veikko Mylläri and Ms. Anita Sassali for participating in
snow impurity sampling.

**Financial support**

The work was financially supported by Academy of Finland in the projects SnowAPP (315497), OPTICA (295874),
Reflectance of Boreal forests (120949), Scattering (260027), Arctic Absorbing Aerosols and Albedo of Snow (A4), (254195),
NABCEA (296302) and by EUMETSAT in the projects Satellite Application Facility on Climate Monitoring (CM SAF),





Satellite Application Facility on Support to Operational Hydrology and Water Management (H SAF) and Satellite Application Facility on Land Surface Analysis (LSA).

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





**Table 1. The variation range and median values of the air temperature and snow surface temperature during the SNORTEX campaign in Sodankylä in March 11 – 19 and April 20 – 27, 2009. The corresponding variation of the snow density and snow water equivalent are given as well. The measurements were carried out during 9 and 17 hours local time (Manninen and Roujean, 2014). The values in brackets are those measured at the reference site NorSEN-mast (67.3621°N, 26.63445°E) in Tähtelä. The total number of individual measurements was altogether 118 (17) for the reference site. The elemental carbon and organic carbon concentrations measured nearby the NorSEN mast (67.364011°N, 26.635891°E) in March and April are shown as well (Meinander et al., 2020).**

| Parameter | | Air temperature at 2 m [°C] | Snow surface temperature [°C] | Snowpack depth [cm] | Snowpack density [g/cm³] | Snowpack water equivalent [mm] | Elemental carbon [ng/g] | Organic carbon [ng/g] |
|---|---|---|---|---|---|---|---|---|
| March | Min | -11.0 (-7.3) | -13.3 (-10.7) | 20 (56) | 0.18 (0.22) | 51 (123) | 18.8 | 483 |
| | Median | -0.4 (0.0) | -2.5 (-4.1) | 57 (59) | 0.23 (0.23) | 129 (139) | 29.4 | 926 |
| | Max | 1.4 (1.2) | 0.45 (0.45) | 71 (62) | 0.29 (0.24) | 225 (149) | 41.5 | 1845 |
| April | Min | -3.5 (-0.05) | -3.6 (-2.2) | 0 (36) | 0.25 (0.22) | 14 (109) | 15.7 | 988 |
| | Median | 5.4 (5.4) | 0.05 (0.15) | 37 (43) | 0.31 (0.27) | 121 (130) | 85.7 | 2894 |
| | Max | 10.13 (8.9) | 0.25 (0.25) | 76 (62) | 0.42 (0.33) | 294 (145) | 106.3 | 7172 |






**Table 2. The variation range and median values of snow grain size (defined as the maximum diameter of the smallest snow grain in each sample) and density of the topmost layer and the snowpack during the SNORTEX campaign in Sodankylä in March 11 – 19 and April 20 – 27, 2009.**

| Parameter | | Grain diameter of top layer [mm] | Grain diameter of snowpack [mm] | Density of top layer [g/cm³] | Density of snowpack [g/cm³] |
|---|---|---|---|---|---|
| March | Min | 0.25 | 0.25 | 0.110 | 0.059 |
| | Median | 0.5 | 1.5 | 0.143 | 0.173 |
| | Max | 1.5 | 3.25 | 0.317 | 0.345 |
| April | Min | 0.25 | 0.25 | 0.065 | 0.011 |
| | Median | 2 | 2 | 0.272 | 0.259 |
| | Max | 3.5 | 4 | 0.433 | 0.433 |





**Table 3. The mean values and variation range of the ratio $r_{fb}$ of backward to forward scattering for diverse solar zenith angle values**
**for all profiles measured in March and April, 2009. These values represent the surface scattering contribution only.**

| Solar zenith angle | Mean $r_{fb}$ | 80% variation range of $r_{fb}$ |
|---|---|---|
| 20° | 0.35 | 0.21 … 0.49 |
| 40° | 0.66 | 0.36 … 1.02 |
| 60° | 1.27 | 0.50 … 2.27 |
| 80° | 4.05 | 1.23 … 8.43 |



**Figure captions:**

**Figure 1.** Test area in Sodankylä in northern Finland. The premises of the Arctic Space Centre of FMI are situated in Tähtelä (T). The operational albedo measurements are located in the upper part of that rectangle, the NorSEN mast at the lower part. The aapa mire test site Mantovaaranaapa is marked with M and the forest clearing site Hirviäkuru with H. The corner co-ordinates of the area given in the WGS84 system. CC BY 4.0 National Land Survey of Finland (04/2020).

**Figure 2.** Examples of surface roughness of snow at Mantovaaranaapa on April 22, 2009. The black background of the
plate is 1 m wide.

**Figure 3.** Facet structure of a randomly rough surface of spherical scatterers.

**Figure 4.** Distributions of the slope angles (in degrees) in March and April calculated from the plate measurements at the average measured spatial resolution of 0.26 mm.

**Figure 5.** The average number of reflections $\langle n_s \rangle$ of individual ray hitting the surface as a function of the rms slope
angle $\beta$ for two incident solar zenith angle values $\theta_i$ for the snow profiles measured in the SNORTEX campaign (Manninen and Roujean 2014; Anttila et al. 2014) in March and April 2009.

**Figure 6.** The average zenith angle $\theta_o$ of reflected escaping individual ray as a function of the rms slope angle $\beta$ for two irradiance solar zenith angle values $\theta_i$ for the snow profiles measured in the SNORTEX campaign (Manninen and Roujean 2014; Anttila et al. 2014) in March and April 2009. The sign of the zenith angle is positive for forward reflection
and negative for backward reflection.

**Figure 7.** Relationship between rms slope and the horizontal distance between the points used for its determination according to the laser scanning data of March 18, 2010. For comparison the mean value and variation range of the rms slope values derived from the 10 plate profiles for horizontal resolution 0.25 mm in the same area in the same day are shown in red. The dashed black curve is the regression to the 36 laser scanning based points (black polyline) and the
grey shaded area covers the 80% variation range of rms slope at the distance in question.

**Figure 8.** The relationship between the albedo values corresponding to the solar zenith angle value of 73° measured operationally at Sodankylä and the regression (albedo = 0.84 - 0.29 $b$ - 0.008 $k_0$) based on measured surface roughness parameters $b$ (Eq. 1) and $k_0$ (Eq. 2) at the NorSEN mast in March and April, 2009. The darkness of the markers is related to the fraction of diffuse irradiance.

**Figure 9.** Variation range (grey) of the snow spectra measured using the ASD spectrometer in Hirviäkuru (67.38°N, 26.85°E) on March 13 and in Mantovaaranaapa (67.4°N, 26.72°E) on April 22. The albedo simulations using the TARTES model are shown for fractal grains (blue) and spheres (red). The solid lines indicate the mean value and the dotted lines the minimum and maximum curves of the day in question. The ASD measurements were carried out in the same area as the grain size and density measurements, but the impurity measurements were daily values measured at
Tähtelä (67.37°N, 26.63°E).

**Figure 10.** Evolution of measured surface albedo in Tähtelä (67.37°N, 26.63°E) on March 12 - 19 and April 20 – 28, 2009. The corresponding variation range of simulated albedo values based on simultaneous grain size and density measurements and temporally interpolated impurity content in the same day are shown in grey. The minimum, mean and maximum values are indicated with darker grey curves. The vertical 'error bars' marked for some of the measured
albedo values are based on the variation range of the measured ASD spectra based broadband reflectance values during the same day in the larger test area in Sodankylä.





**Figure 11. Ratio of simulated total and bulk white-sky albedo values for all 1381 profiles and snow pit and impurity data in Sodankylä in March 11-19 and April 20-28, 2009. The daily mean, minimum and maximum values are indicated by the darker curve and the variation range is shaded.**

**Figure 12. Simulated albedo values at the NorSEN-mast based on density, grain size and surface roughness measurements of snow in March 12-19, 2009 and in April 20 – 28, 2009 and temporally interpolated impurity content data. The mean values are shown separately for the TARTES model containing only the volume scattering contribution (grey curve) and the TARTES model combined with the surface scattering model of this study (blue or red). The daily variation range based on diverse profiles, grain size and density measured at the site and temporally interpolated**
**impurity content is shown as the area shaded by light blue or light red colour. The darkness of the empirical points indicates the fraction of diffuse irradiance during the measurement.**

**Figure 13. Measured and modelled blue-sky albedo values at Mantovaaranaapa on April 22, 2009. Each modelled point is an average corresponding to three individual plate profiles taken from the same surface. The empirical albedo values are likewise averages of individual three points recorded using the Kipp & Zonen albedometer CM14 at the time of**
**taking the profile photos. The shaded area shows the variation range of the broadband reflectance values measured using the ASD spectrometer.**

**Figure 14. Measured principal plane BRFs in Mantovaaranaapa April 22, 2009 of one smooth snow and three rough snow cases.**

**Figure 15. Mantovaaranaapa on April 22, 2009.**

**Figure 16. Calculated average scattering angle distributions (i.e., zenith angles of reflected radiation, positive for forward directions and negative for backward directions) for profiles measured during the SNORTEX campaign in 2009 (blue) and in Mantovaaranaapa in April 22, 2009 (red) for incidence angle values 20° and 60°.**

**Figure 17. Surface roughness parameter *a* as a function of surface roughness parameter *b* (Eq. 1) of the dominantly backscattering and dominantly forward scattering profiles for incidence angles 20°, 40° and 60°.**

**Figure 18. Snow surface structure evolution during March 16 - 18, 2009. The cm scale is shown in the right images.**

**Figure 19. Microscale snow surface structure evolution during March 16 - 18, 2009. In the left images the scale of the grid is 1 mm. In the right images the grids correspond to 1 mm, 2 mm and 3 mm from left to right.**

**Figure 20. Top) The snow surface rms height and correlation length measured with the plate method during March 16 – 18, 2009. The values correspond to the distance 60 cm. Bottom) The rms slope $\beta$ of individual plate snow profiles**
**during March 16 – 18, 2009. The linear regression for $\beta$ vs. time is shown for March 16, $R^2$ value included.**





(67.45°N, 26.87°E)

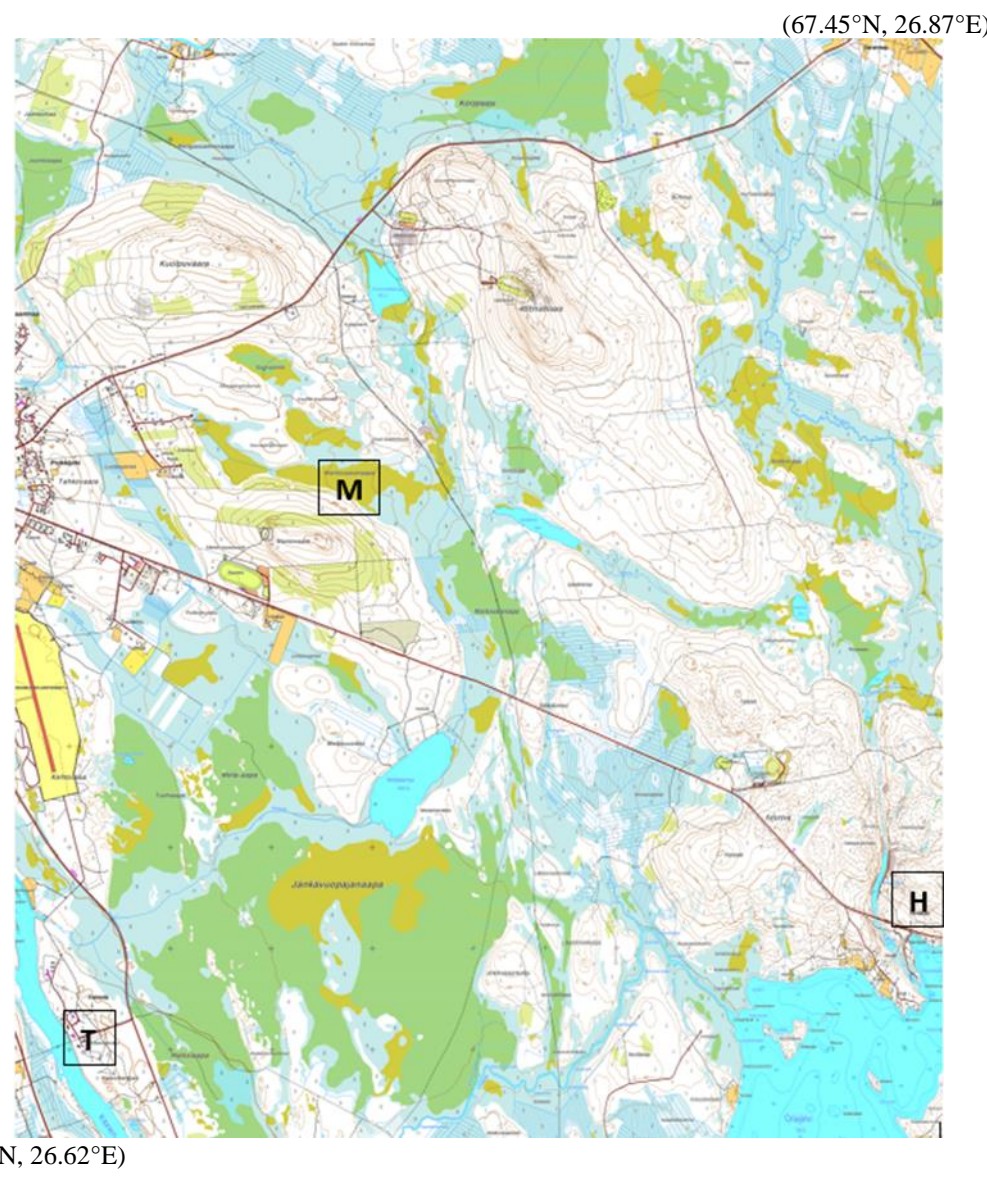

(67.35°N, 26.62°E)


Figure 1. Test area in Sodankylä in northern Finland. The premises of the Arctic Space Centre of FMI are situated in Tähtelä
(T). The operational albedo measurements are located in the upper part of that rectangle, the NorSEN mast at the lower part.
The aapa mire test site Mantovaaranaapa is marked with M and the forest clearing site Hirviäkuru with H. The corner co-
ordinates of the area given in the WGS84 system. CC BY 4.0 National Land Survey of Finland (04/2020).





Figure 2. Examples of surface roughness of snow at Mantovaaranaapa on April 22, 2009. The black background of the plate is 1 m wide.





Figure 3. Facet structure of a randomly rough surface of spherical scatterers.





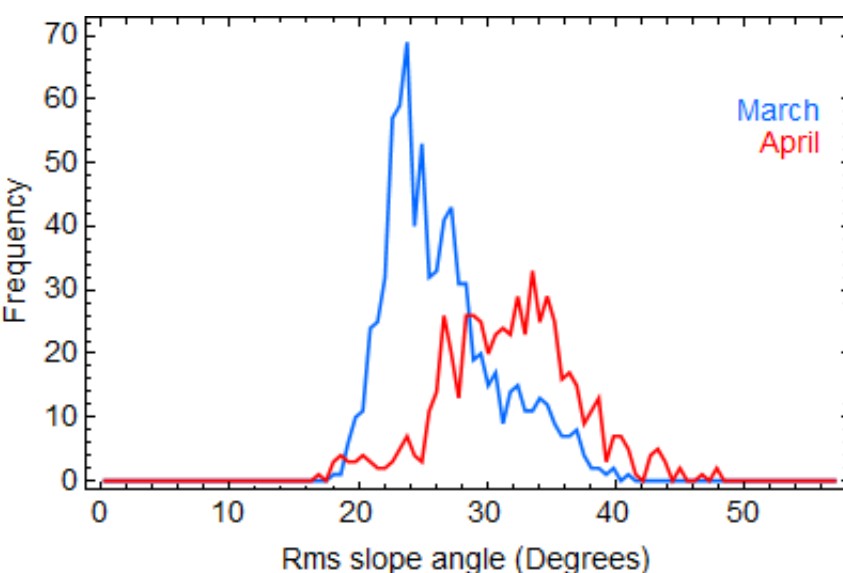


Figure 4. Distributions of the slope angles (in degrees) in March and April calculated from the plate measurements at the average measured spatial resolution of 0.26 mm.






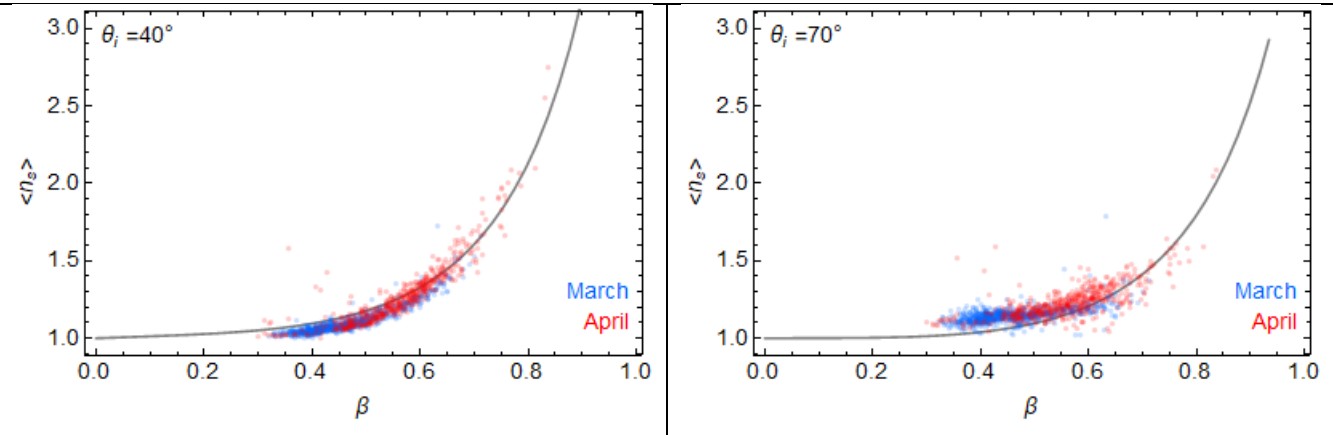


Figure 5. The average number of reflections $<n_s>$ of individual ray hitting the surface as a function of the rms slope angle $\beta$ for two incident solar zenith angle values $\theta_i$ for the snow profiles measured in the SNORTEX campaign (Manninen and Roujean 2014; Anttila et al. 2014) in March and April 2009.






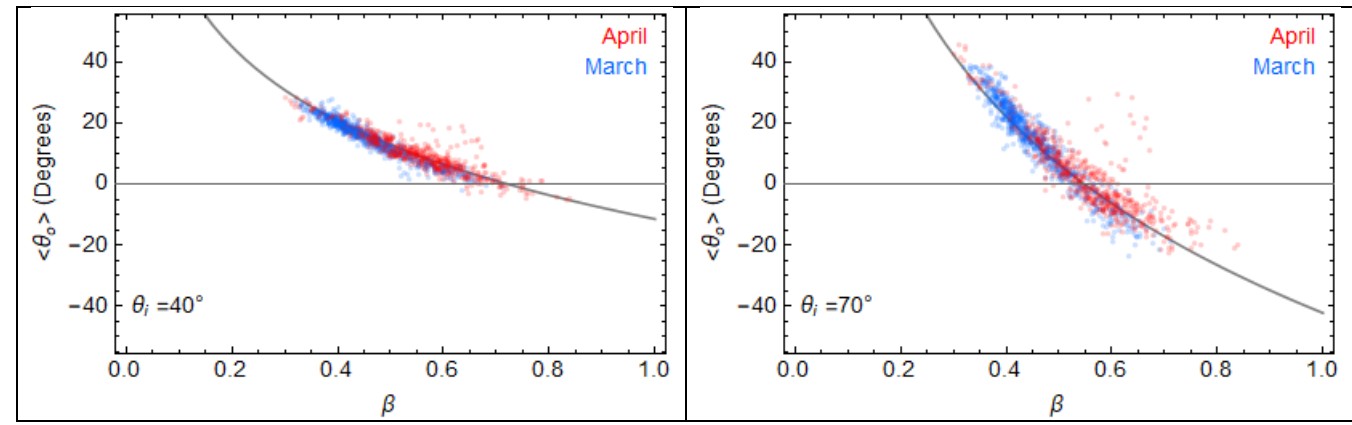


Figure 6. The average zenith angle $\theta_o$ of reflected escaping individual ray as a function of the rms slope angle $\beta$ for two irradiance solar zenith angle values $\theta_i$ for the snow profiles measured in the SNORTEX campaign (Manninen and Roujean 2014; Anttila et al. 2014) in March and April 2009. The sign of the zenith angle is positive for forward reflection and negative for backward reflection.




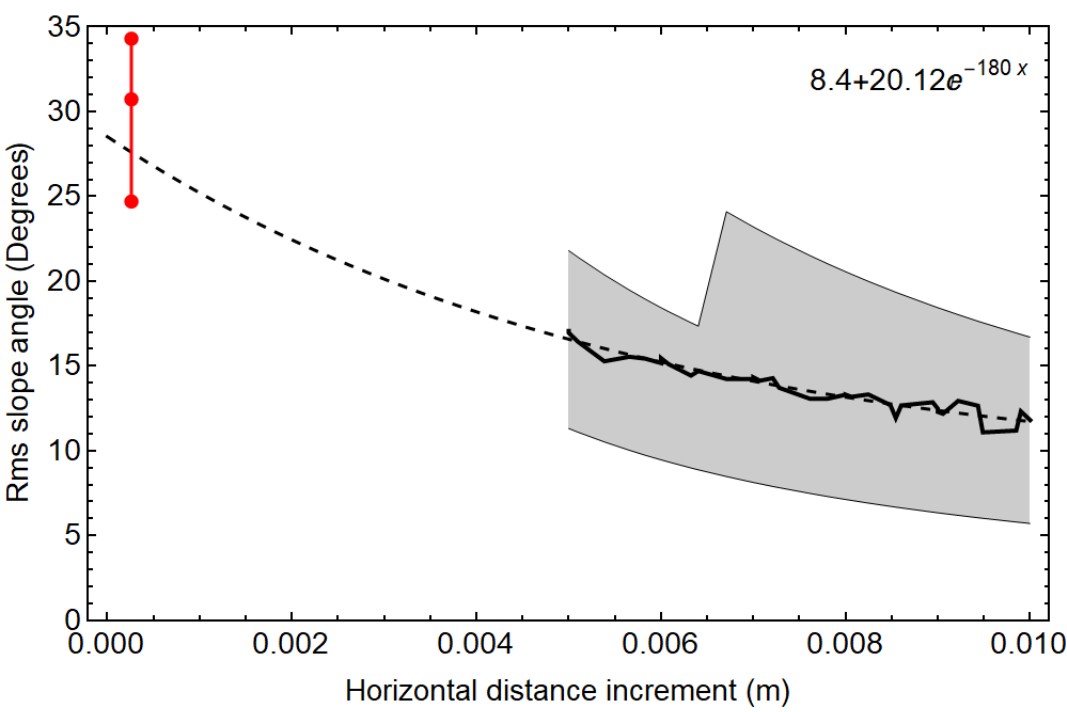


Figure 7. Relationship between rms slope and the horizontal distance between the points used for its determination according to the laser scanning data of March 18, 2010. For comparison the mean value and variation range of the rms slope values derived from the 10 plate profiles for horizontal resolution 0.25 mm in the same area in the same day are shown in red. The dashed black curve is the regression to the 36 laser scanning based points (black polyline) and the grey shaded area covers

the 80% variation range of rms slope at the distance in question.



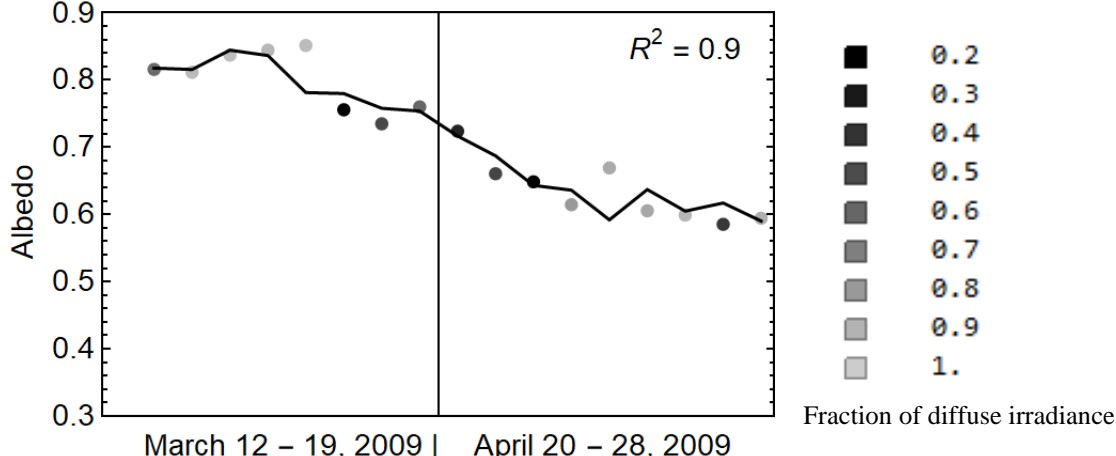

Figure 8. The relationship between the albedo values corresponding to the solar zenith angle value of 73° measured operationally at Sodankylä and the regression (albedo = 0.84 - 0.29 *b* - 0.008 *k0*) based on measured surface roughness parameters *b* (Eq. 1) and *k0* (Eq. 2) at the NorSEN mast in March and April, 2009. The darkness of the markers is related to the fraction of diffuse irradiance.



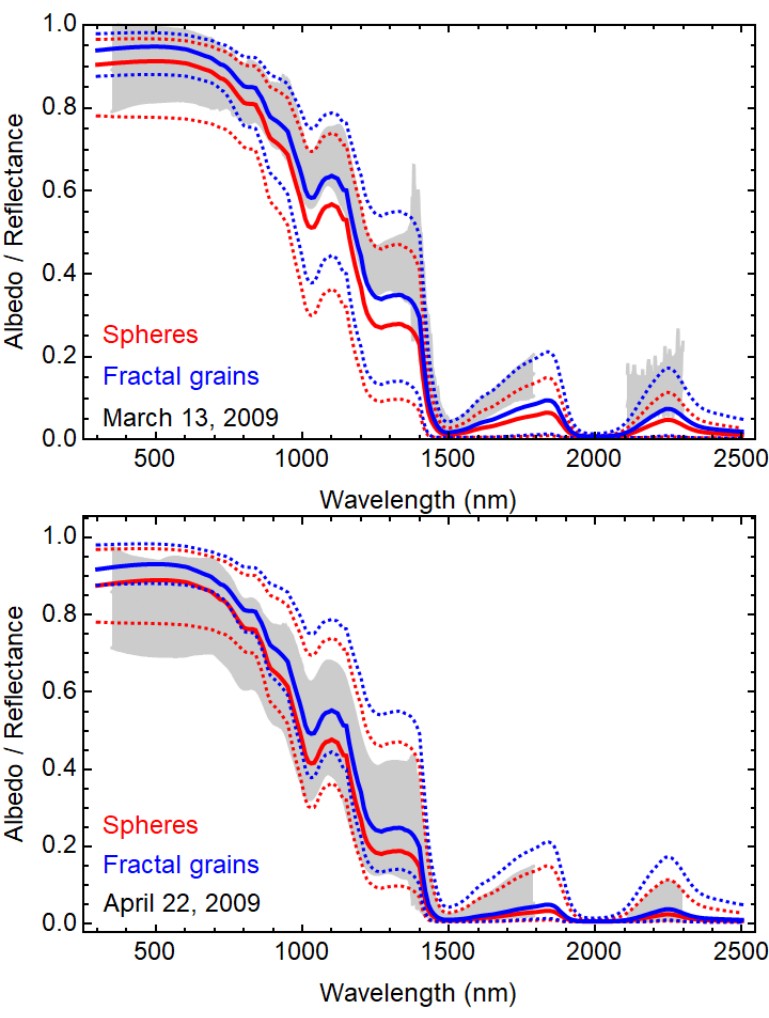

Figure 9. Variation range (grey) of the snow spectra measured using the ASD spectrometer in Hirviäkuru (67.38°N, 26.85°E) on March 13 and in Mantovaaranaapa (67.4°N, 26.72°E) on April 22. The albedo simulations using the TARTES model are shown for fractal grains (blue) and spheres (red). The solid lines indicate the mean value and the dotted lines the minimum and maximum curves of the day in question. The ASD measurements were carried out in the same area as the grain size and density measurements, but the impurity measurements were daily values measured at Tähtelä (67.37°N, 26.63°E).





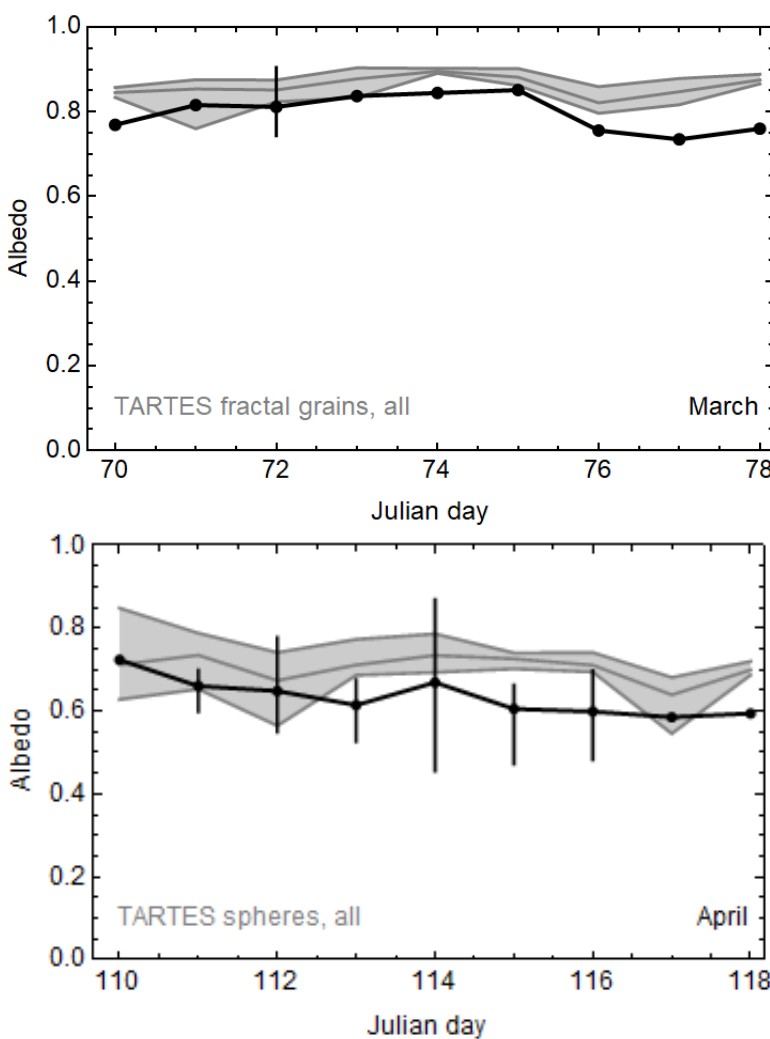

Figure 10. Evolution of measured surface albedo in Tähtelä (67.37°N, 26.63°E) on March 12 - 19 and April 20 – 28, 2009.
The corresponding variation range of simulated albedo values based on simultaneous grain size and density measurements
and temporally interpolated impurity content in the same day are shown in grey. The minimum, mean and maximum values
are indicated with darker grey curves. The vertical 'error bars' marked for some of the measured albedo values are based on
the variation range of the measured ASD spectra based broadband reflectance values during the same day in the larger test
area in Sodankylä.





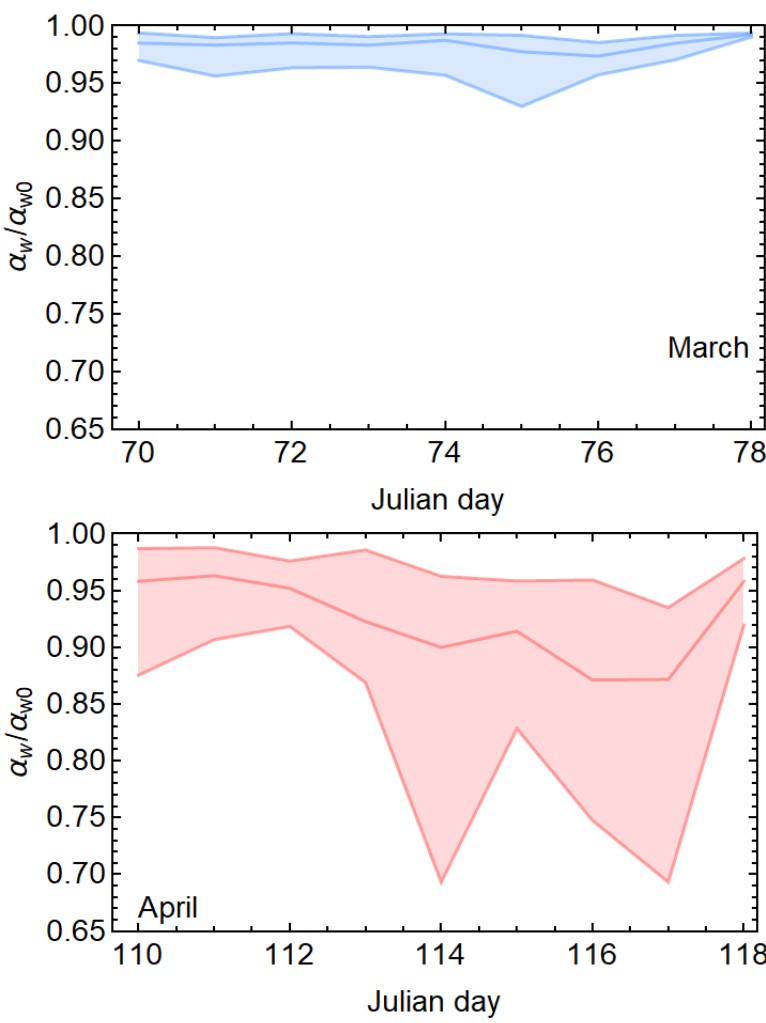

Figure 11. Ratio of simulated total and bulk white-sky albedo values for all 1381 profiles and snow pit and impurity data in Sodankylä in March 11-19 and April 20-28, 2009. The daily mean, minimum and maximum values are indicated by the darker curve and the variation range is shaded.






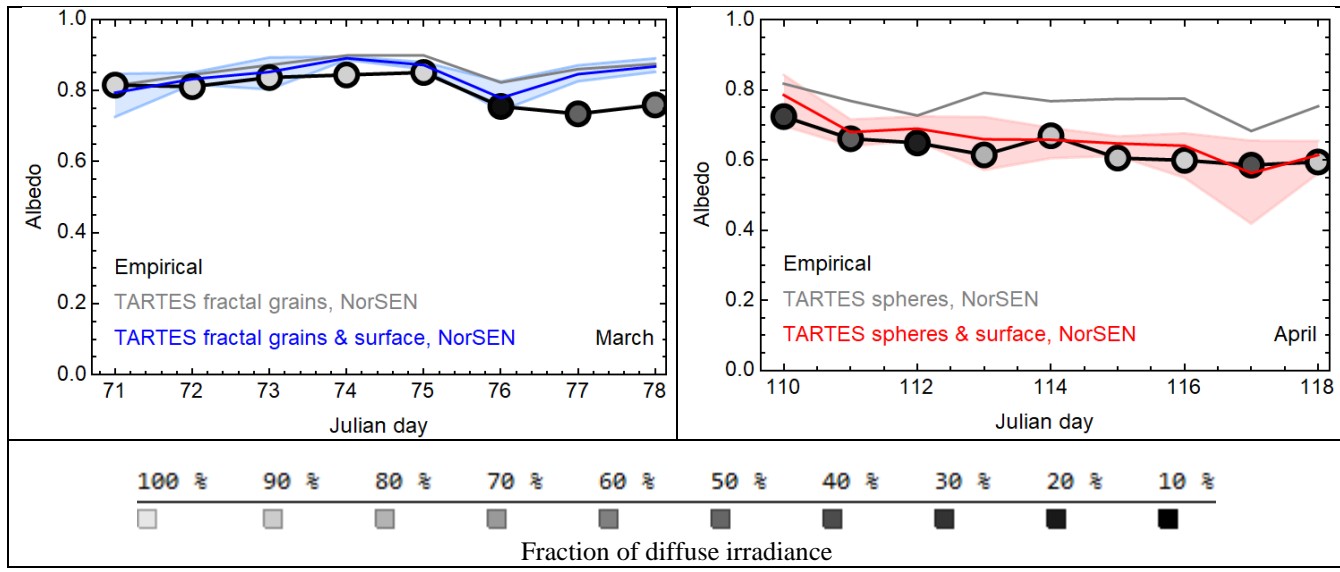

Figure 12. Simulated albedo values at the NorSEN-mast based on density, grain size and surface roughness measurements of snow in March 12-19, 2009 and in April 20 – 28, 2009 and temporally interpolated impurity content data. The mean values are shown separately for the TARTES model containing only the volume scattering contribution (grey curve) and the TARTES model combined with the surface scattering model of this study (blue or red). The daily variation range based on diverse profiles, grain size and density measured at the site and temporally interpolated impurity content is shown as the area shaded by light blue or light red colour. The darkness of the empirical points indicates the fraction of diffuse irradiance during the measurement.





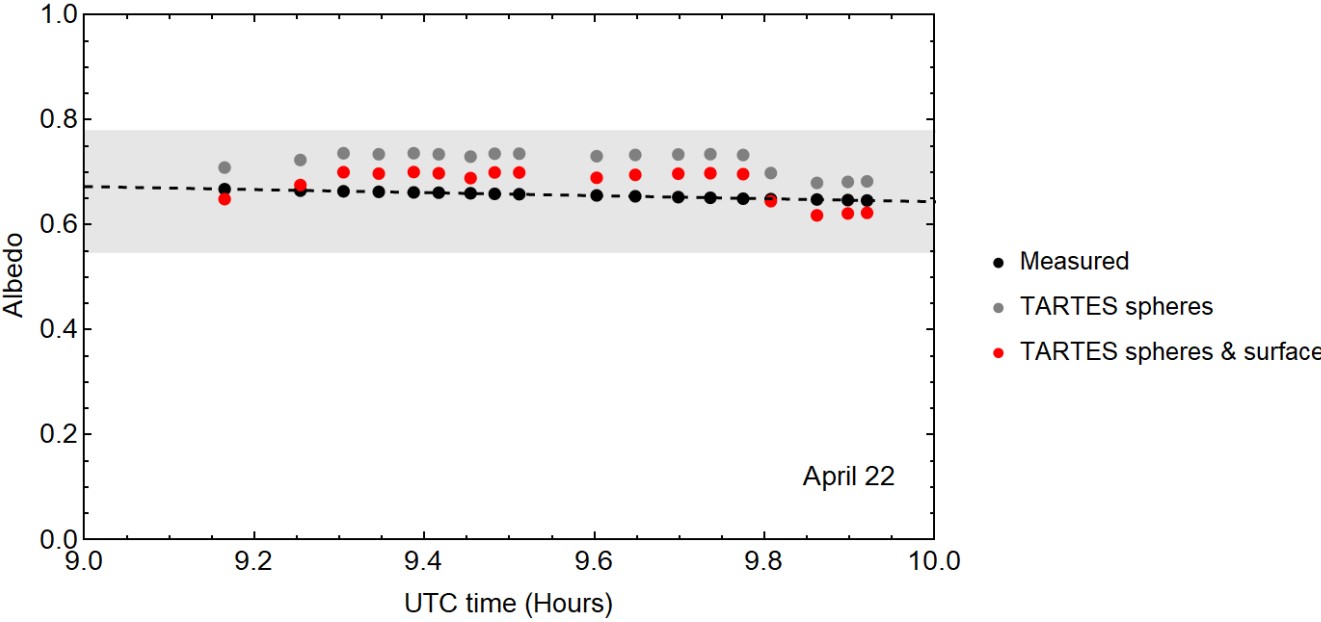

Figure 13. Measured and modelled blue-sky albedo values at Mantovaaranaapa on April 22, 2009. Each modelled point is an
average corresponding to three individual plate profiles taken from the same surface. The empirical albedo values are
likewise averages of individual three points recorded using the Kipp & Zonen albedometer CM14 at the time of taking the
profile photos. The shaded area shows the variation range of the broadband reflectance values measured using the ASD
spectrometer.





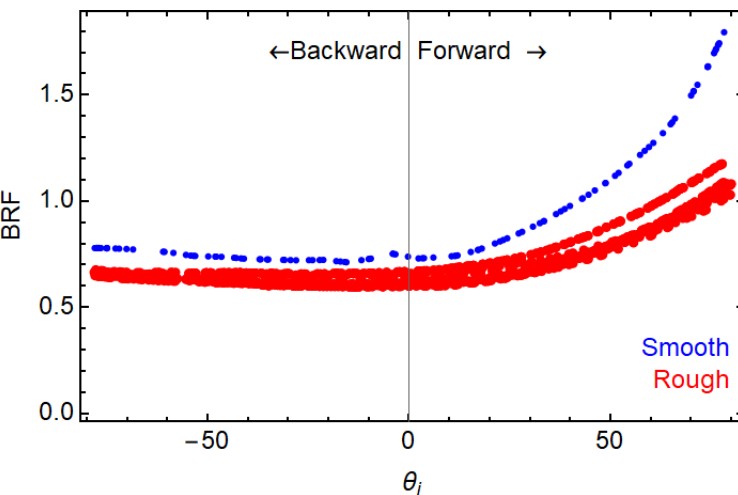


Figure 14. Measured principal plane BRFs in Mantovaaranaapa April 22, 2009 of one smooth snow and three rough snow cases.



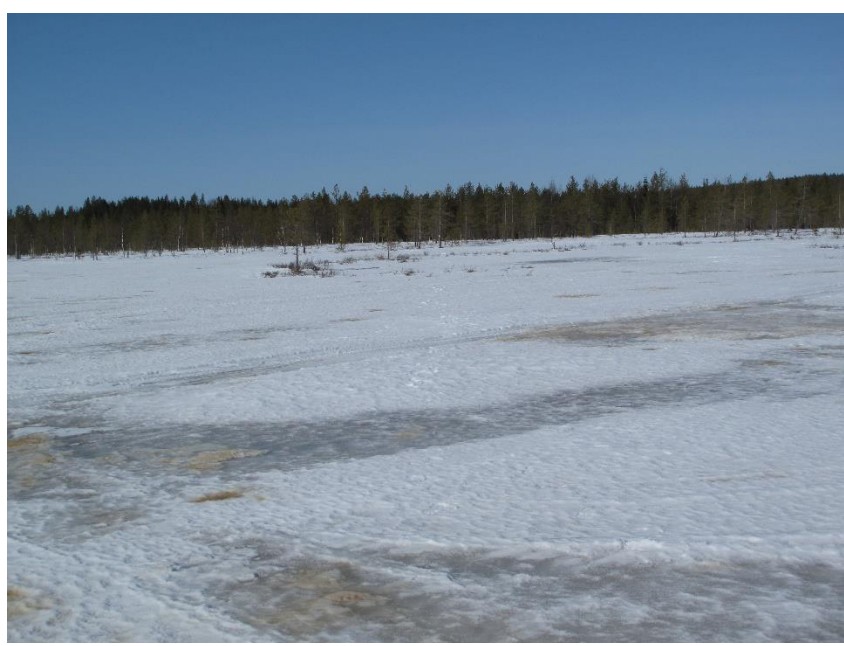


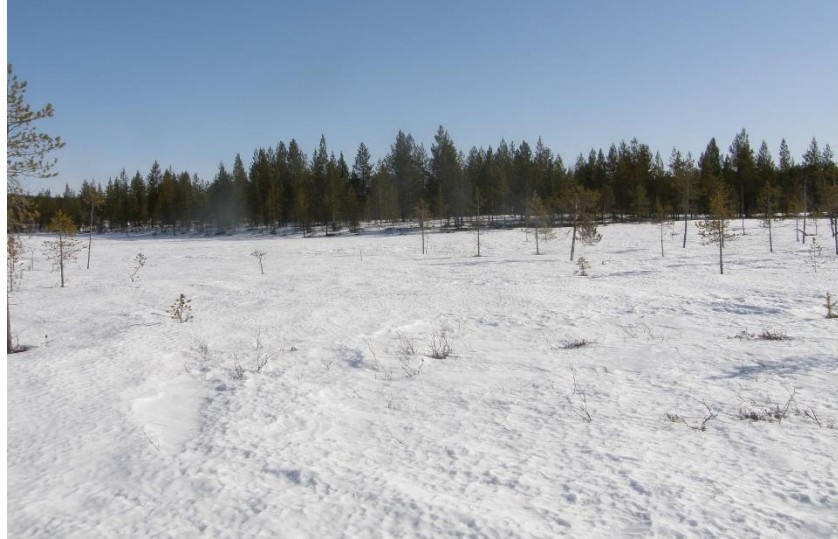

Figure 15. Mantovaaranaapa on April 22, 2009.






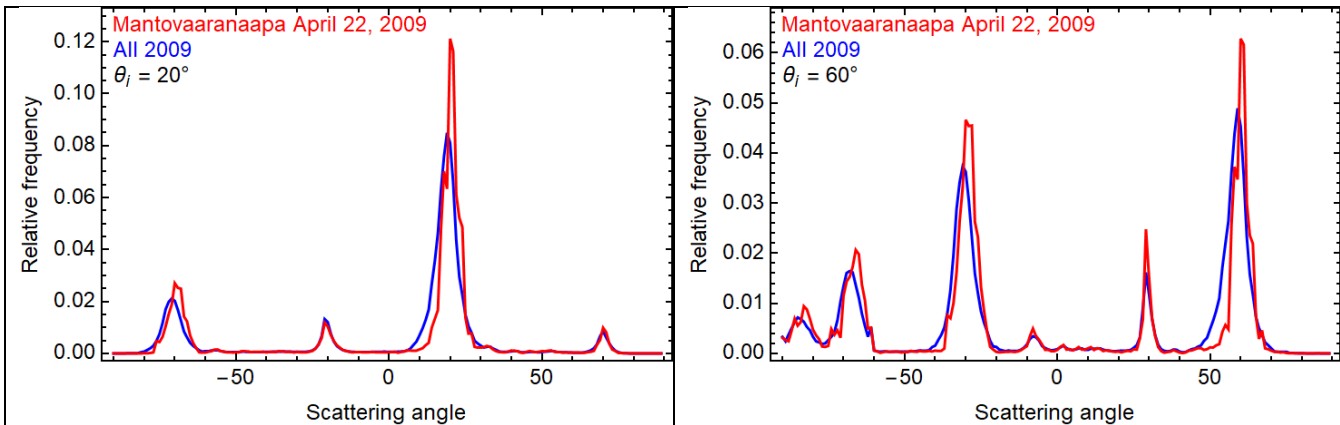

Figure 16. Calculated average scattering angle distributions (i.e., zenith angles of reflected radiation, positive for forward directions and negative for backward directions) for profiles measured during the SNORTEX campaign in 2009 (blue) and in Mantovaaranaapa in April 22, 2009 (red) for incidence angle values 20° and 60°.






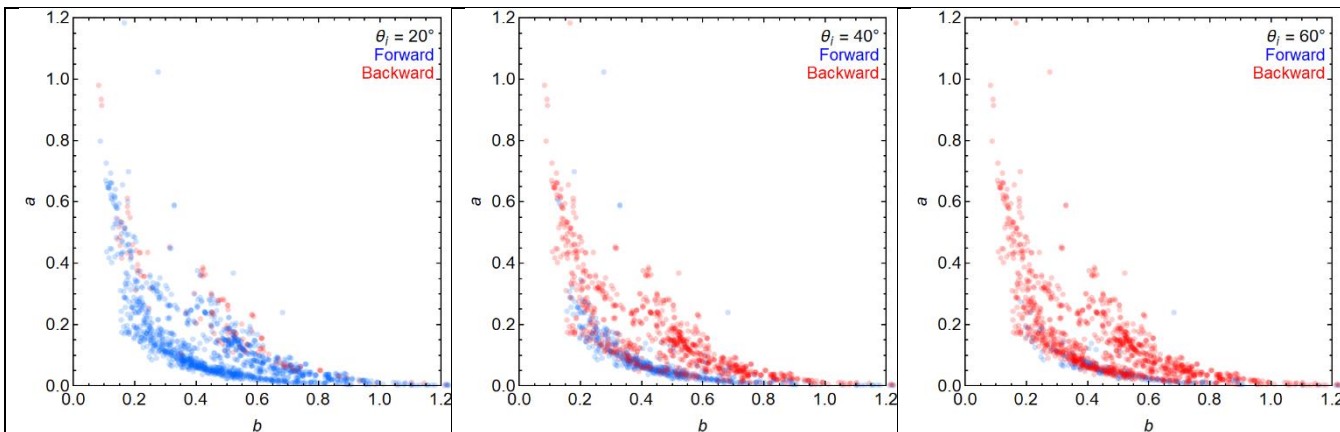

Figure 17. Surface roughness parameter *a* as a function of surface roughness parameter *b* (Eq. 1) of the dominantly backscattering and dominantly forward scattering profiles for incidence angles 20°, 40°and 60°.



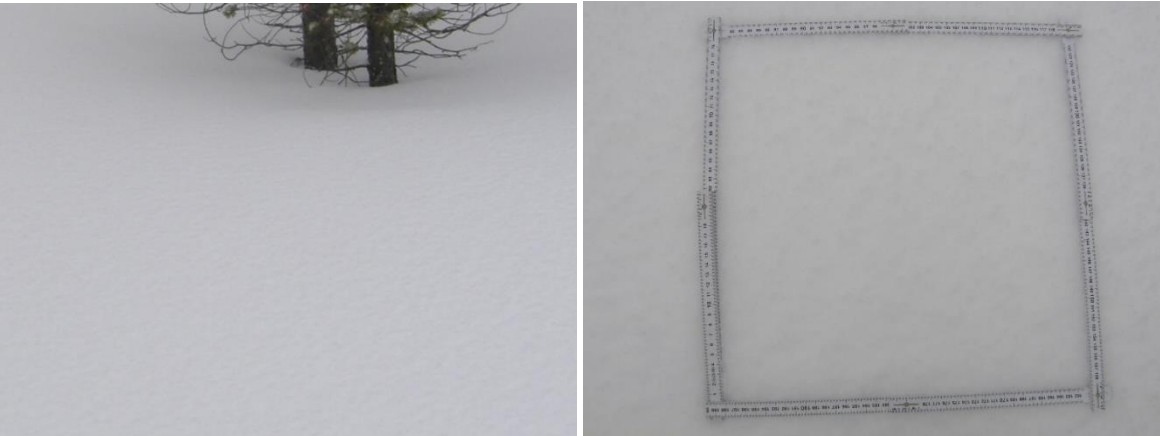

March 16, 2009

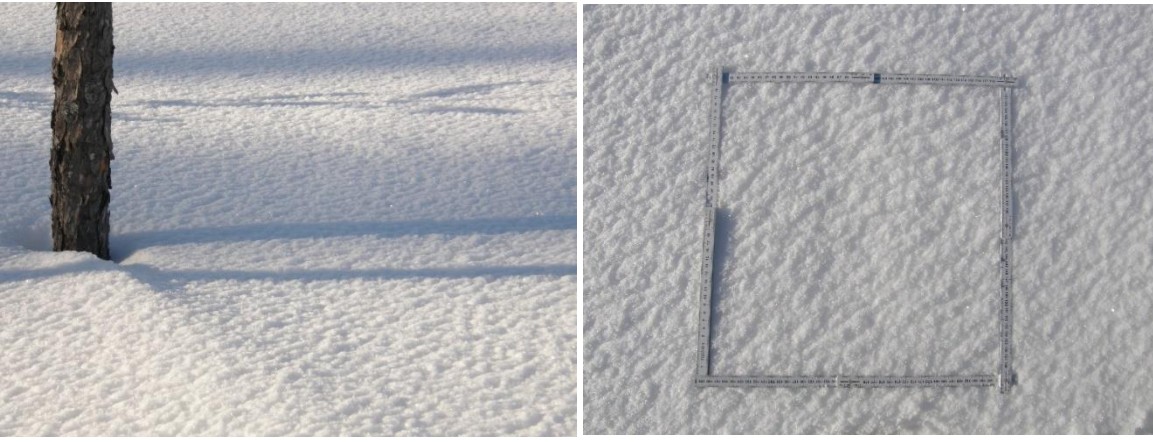

March 17, 2009

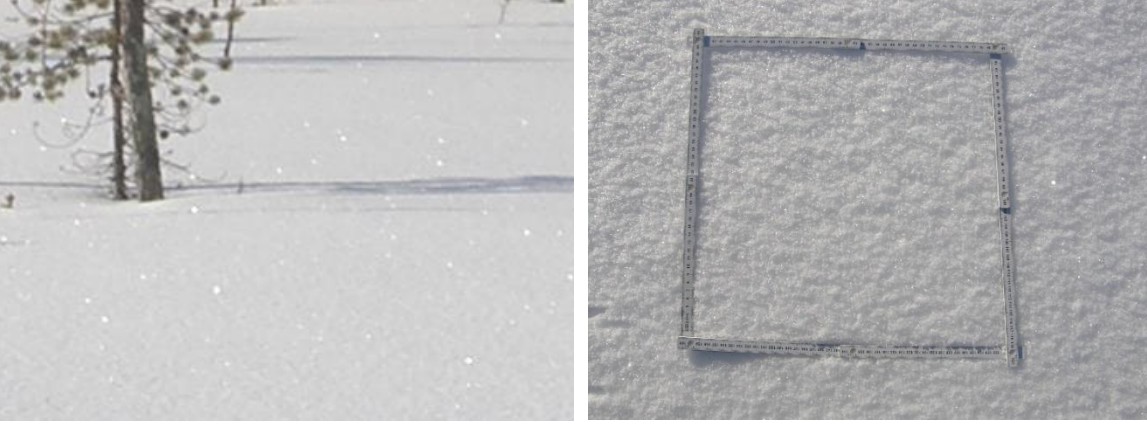

March 18, 2009

Figure 18. Snow surface structure evolution during March 16 - 18, 2009. The cm scale is shown in the right images.



March 16

March 17

March 18

Figure 19. Microscale snow surface structure evolution during March 16 - 18, 2009. In the left images the scale of the grid is 1 mm. In the right images the grids correspond to 1 mm, 2 mm and 3 mm from left to right.


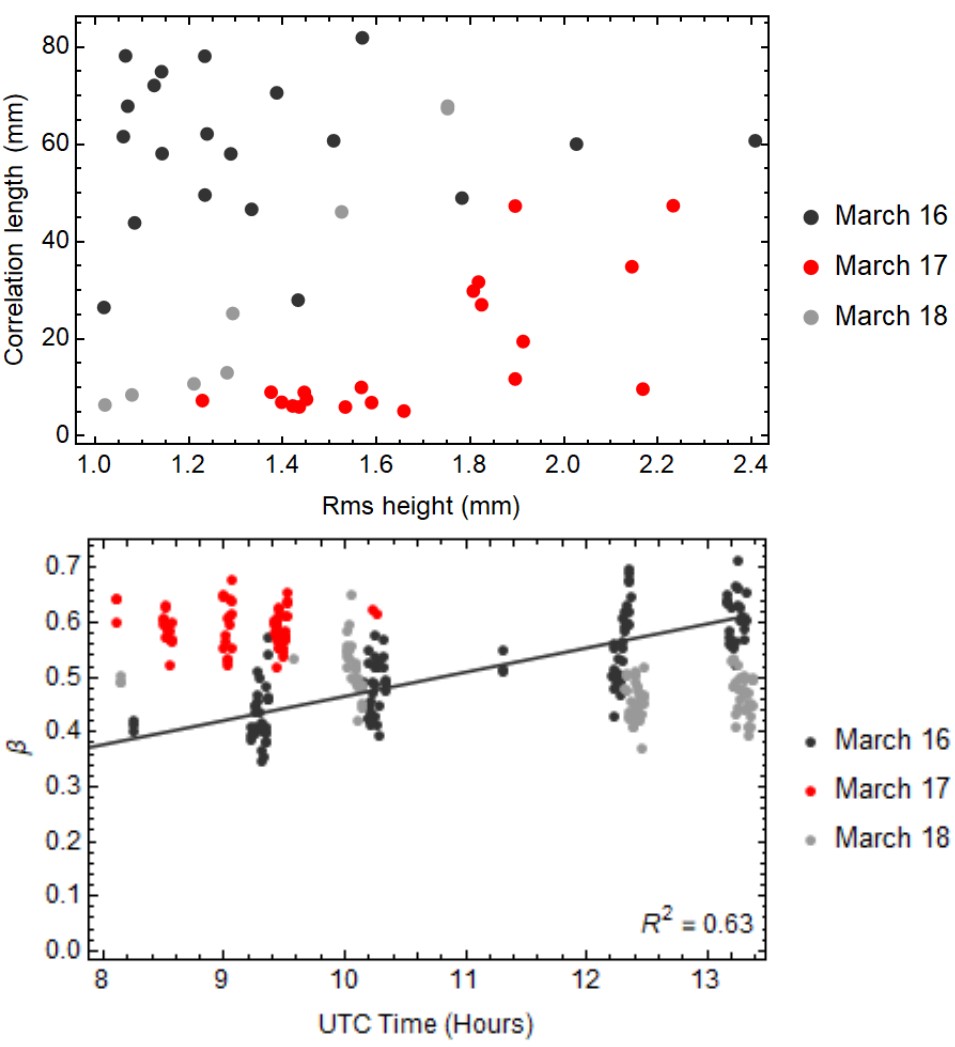

Figure 20. Top) The snow surface rms height and correlation length measured with the plate method during March 16 – 18, 2009. The values correspond to the distance 60 cm. Bottom) The rms slope $\beta$ of individual plate snow profiles during March 16 – 18, 2009. The linear regression for $\beta$ vs. time is shown for March 16, $R^2$ value included.