# Peer review of "Effect of small-scale snow surface roughness on snow albedo and reflectance"

_The Cryosphere, 2020_

## Referee Comment (RC1) · Alexander Kokhanovsky (Referee) · 15 Sep 2020

The reflection of light from snow surfaces is inflenced by light scattering within snow layer and also by snow surface roughness. Usually the snow BRDF is studied using the radiative transfer theory for plane - parallel media and effects of surface roughness are ignored. The authors of this work propose a technique, which can be used to study the efefct of small-scale snow surface roughness on snow albedo and reflectance. The paper is sound and can be published subject to minor revisions. My comments are as follows: 1). line 60: snowpack, 2)Eqs. 1, and 2: please, give more explanation on the meaning of x and L, 3), 3) line 166, remove '.', 4) Eq. (7): define all variables, 5) please, check out the style of references ( see, e.g., p.27), 6) Fig.9: please, explain the minimum around 500nm for the case of April 22, 2009, 7) Figs. 8, 13: please, write

'broadband albedo' and not albedo along axis OY, 8) Please, improve figure captions (e.g., you need to give units in Figs.14, 16).

---

## Referee Comment (RC3) · Anonymous Referee #2 · 6 Nov 2020

The authors presented a snow surface albedo model with small-scale surface roughness being considered. The model was constructed by combining the bulk volume albedo model (TARTES) with a surface scattering model based on the photon recollision probability theory. Overall, the manuscript is well written. Nevertheless, I agree with Picard that the authors should discuss their results with respect to the recent literature like Larue et al. 2020 to further improve the manuscript.

Detailed comments:

Line 131: rms -> root mean square?

Line 131-141: This section is not clear. Elaborate more about "the rms height and correlation and their distance dependence". It would be helpful if the authors can make

a schematic diagram to illustrate the parameters and variables (the distance x and the correlation length L) in equation (1) and (2). In addition, explain "The snow surface roughness is close to a Brownian fractal surface" with more details.

Line 144: "The vertical resolution was about 0.1 mm and the horizontal resolution 0.04 mm." These two resolutions, you mean precision? It is a bit confusing with the context ". . .calculated for the measured spatial resolution, which was on the average 0.26 mm."

Line 150-156: what is the exact data acquisition precision for your dataset? How does it compare with the plate measurement precision and resolution?

Line 158-160: Is it possible to provide a figure to show the snowmobile tracks? and one scanned profile? How was "the root mean square (rms) slopes per size of horizontal distance increment" calculated? What's the unit?

Line 209: per cent -> percent

Line 212: "complemented by set or artificial light measurements", typos?

Equation (4): please make the equation and the description of variables consistent.

Line 313-314: provide the surface height distribution histograms like figure 4.

Equation (7) and (8): give a detailed description of each variable shown in the equations.

Line 356-369: This section involves lots of statistical correlation analyses between surface roughness parameters and surface albedo measurements. Please provide the scatter plots as supplementary material to show the correlations. The authors used the rms height as the only explanatory variable to explain the variability of albedo and obtained quite high R square and therefore concluded that "This result supports the view that surface roughness affects the albedo." I don't think this is a solid statement given that the snow grain size variations from March to April could greatly affect the variability of surface albedo. Without controlling for the snow grain size, it is not plausible to validate the relationship between rms slope and surface albedo. The authors should take both rms slope and snow grain size as the explanatory variables to explain the variability of surface albedo, and then compare the relative contributions of rms slope and snow grain size.

Discussion: according to the analysis, it appears that the spatial scale matters when evaluating the impact of surface roughness to snow albedo. Based on the results, can the authors discuss the implications of this study for studying the snow albedo changes using satellite data of different resolutions? Is there a rough estimate on how much surface roughness would affect the satellite albedo (such as MODIS albedo)?

Figure 3. Please label the figure to indicate the meaning of each different line.

Figure 11. For the April data, it appears that the uncertainty was dramatically increased after Julian day 112, any possible reason?

Figure 13. Shrink the Y axis scale to 0.5~0.8 instead of using 0-1, otherwise, the measured albedo seems to be constant through the time.

Figure 14. Are there three different profiles for the "rough" surface? If so, label the rough profiles differently (different parameters)?

---

## Author Comment (AC4) · 25 Nov 2020

Supplementary material for the manuscript " Effect of small-scale snow surface roughness on snow albedo and reflectance" by Terhikki Manninen, Kati Anttila, Emmihenna Jääskeläinen, Aku Riihelä, Jouni Peltoniemi, Petri Räisänen, Panu Lahtinen, Niilo Siljamo, Laura Thölix, Outi Meinander, Anna Kontu, Hanne Suokanerva, Roberta Pirazzini, Juha Suomalainen, Teemu Hakala, Sanna Kaasalainen, Harri Kaartinen, Antero Kukko, Olivier Hautecoeur and Jean-Louis Roujean

The figures are related to the text starting in line 376:

The relationship between other surface roughness parameters (such as rms height $\sigma$ and correlation length $L$) and $\beta$ is in general not strong even for a Gaussian surface height distribution (Beckmann & Spizzicchino, 1963). For the whole period (March 3 – April 28, 2009) the ratio of $\sigma/L$ (determined for 0.60 m distance) correlated however relatively well with $\beta$, the $R^2$ values being 0.70 (**Figure. S1**), 0.62 and 0.67 for the whole data range, March and April, respectively, but the best descriptor of $\beta$ was found to be $\sigma/b$ (Eq. 1, **Figure. S2**), its $R^2$ values for the linear correlation being 0.78, 0.68 and 0.82 for the whole data range, March and April, respectively. $\beta$ tends to increase with the progress of the melting season (0.002 radians per day). Likewise, its correlation with $\sigma/b$ increases during the melting season. It was therefore examined, whether the measured surface albedo correlates well with the measured surface roughness parameters. Using just the rms height (derived for a 0.60 m horizontal scale) as an explanatory variable of the albedo the coefficient of determination was $R^2 = 0.81$ (**Figure. S3**). The relationship between the albedo and surface roughness parameters that are scale-independent in a large range (Manninen, 2003) was then evaluated. Indeed, a simple linear regression for the data of March and April, 2009 produced a coefficient of determination value as high as $R^2 = 0.90$, when the parameters $b$ and $k_0$ (see Eqs. 1 and 2) were used as explanatory variables (Figure 8). While correlation is not a proof of causality, this result supports the view that surface roughness affects the albedo.

[Figure]

Figure S1. Rms slope (in radians) vs. the ratio of the average daily rms height and the average daily correlation length in March and in April.

[Figure]

Figure S2. Rms slope (in radians) vs. the ratio of the average daily rms height and the average daily roughness parameter b of Eq.1 in March and in April.

[Figure]

Figure S3. Albedo as a function of the daily average rms height.

---

## Author Response (AR1)

Answers to reviewer Alexander Kokhanovsky:

Reviewers comments are shown in black, authors' answers in blue, and there the line numbers refer to the revised version.

The reflection of light from snow surfaces is inflenced by light scattering within snow layer and also by snow surface roughness. Usually the snow BRDF is studied using the radiative transfer theory for plane - parallel media and effects of surface roughness are ignored. The authors of this work propose a technique, which can be used to study the efefct of small-scale snow surface roughness on snow albedo and reflectance. The paper is sound and can be published subject to minor revisions.

My comments are as follows:

1). line 60: snowpack
Edited as suggested.

2) Eqs. 1, and 2: please, give more explanation on the meaning of x and L
The correlation length L is commonly used as one descriptor of surface roughness. And x stands for the length of the analyzed profile. It is intuitively clear that if one wants to describe the roughness of 1 m, 10 m and 100 m long profiles, one will get different numerical values for the roughness parameters, such as rms height, as no natural (or man-made either) surfaces are stationary. The text is slightly edited to make it clear that x is the distance for which the surface roughness parameters $\sigma$ and L are calculated. This roughness analysis is more thoroughly already explained in the given references (Manninen, Physica 2003; Anttila et al., JGR 2014), hence repeating it again in an already long paper does not seem justified, but additional fundamental references are now included (Keller et al., 1987; Church, 1988) in line 134 and the text is slightly modified to help the reader to find this information.

3) line 166, remove '.'
Edited as suggested.

4) Eq. (7): define all variables
The missing definition of $\theta_i$ was added after Eq. 7. Also $\theta_o$ was added in the text.

5) please, check out the style of references ( see, e.g., p.27)
The style of the references was checked and noticed discrepancies were corrected.

6) Fig.9: please, explain the minimum around 500nm for the case of April 22, 2009
The seeming minimum around 500 nm is a result of showing the minimum and maximum of the individual reflectance curves of 15 snow patches as one gray band. One increasing (dirty snow) and one decreasing curve just happen to cross at about 500 nm, see figure below.

[Figure]

However, after all the authors consider the ASD spectrometer measured reflectance values in the UV wavelengths in clear sky conditions unreliable. This results in increasing reflectance with decreasing wavelength at the UV range in the data of Fig. 9b. For very wet or dirty snow or vegetation this effect does not appear, nor for highly reflecting snow in cloudy conditions. Hence, the authors decided to remove the values measured below 500 nm (gray shaded area) from Fig 9b. In Fig. 9a below 500 nm values will remain. The best quality UV albedo measurements on 22 April 2009 clear sky conditions were performed using a Bentham spectrometer in one location besides an open field (Fig. 3 and Fig. 4 of Meinander et al 2013). In that paper the spectral albedo was discussed in detail for 300 -550 nm. The results of Meinander et al. (2013) do not support the general increase of albedo with decreasing wavelength suggested by the ASD spectrometer (the figure above). Rather, it was found that the spectral albedo generally decreased with decreasing wavelength in the 300-550 nm, and it also decreased during the day. These features can be ascribed to the increase of snow grain diameter during the day (from 0.25 to 3 mm; Table 2 of Meinander et al. 2013) and the effect of snow impurities (87 ppb black carbon and 2894 ppb organic carbon). These findings were consistent with the theoretical findings by Warren and Wiscombe (1980).

7) Figs. 8, 13: please, write 'broadband albedo' and not albedo along axis OY
Edited as suggested. Figure captions 10 and 12 were edited likewise.

8) Please, improve figure captions (e.g., you need to give units in Figs.14, 16).

The missing units were added to the labels of the axis of the figures.

Answers to Reviewer #2

Reviewers comments are shown in black, authors' answers in blue, and there the line numbers refer to the revised version.

The authors presented a snow surface albedo model with small-scale surface roughness being considered. The model was constructed by combining the bulk volume albedo model (TARTES) with a surface scattering model based on the photon recollision probability theory. Overall, the manuscript is well written. Nevertheless, I agree with Picard that the authors should discuss their results with respect to the recent literature like Larue et al. 2020 to further improve the manuscript.

The authors definitely agree with the reviewer that the paper by Larue et al. 2020 should be discussed. This paper was not available, when writing of the manuscript was started, but this paper is really interesting and is now added to the list of references and assessed in the introduction and discussion.

Detailed comments:

Line 131: rms -> root mean square?
Yes, this clarification is now added in the text in line 133.

Line 131-141: This section is not clear. Elaborate more about "the rms height and correlation and their distance dependence". It would be helpful if the authors can make a schematic diagram to illustrate the parameters and variables (the distance x and the correlation length L) in equation (1) and (2). In addition, explain "The snow surface roughness is close to a Brownian fractal surface" with more details.
The correlation length L is one of the basic descriptors of surface roughness. And x stands for the length of the analyzed profile. It is intuitively clear that if one wants to describe the roughness of 1 m, 10 m and 100 m long profiles, one will get different numerical values for the roughness parameters, such as rms height, as no natural (or man-made either) surfaces are stationary. The text is slightly edited to make it clear that x is the distance for which the surface roughness parameters $\sigma$ and L are calculated. However, this roughness analysis is already more thoroughly explained in the given references (Manninen, Physica 2003; Anttila et al., JGR 2014), hence repeating it again (including diagrams) in an already long paper does not seem justified, but additional fundamental references are now included in line 134 (Keller et al., 1987; Church, 1988) and the text is slightly modified to help the reader to find this information.

The Brownian fractal surface is a fundamental term in surface roughness description not introduced for the first time in this manuscript. It is governed by Eqs. 1 and 2 already given in the manuscript. If the surface in question obeys those equations, it is Brownian fractal. However, the authors added a basic reference for Brownian fractal surfaces (Russ, 1994) in line 142 for an interested reader.

Line 144: "The vertical resolution was about 0.1 mm and the horizontal resolution 0.04mm." These two resolutions, you mean precision? It is a bit confusing with the context"...calculated for the measured spatial resolution, which was on the average 0.26 mm."
Yes, the term resolution is now replaced by precision in line 145.

Line 150-156: what is the exact data acquisition precision for your dataset? How does it compare with the plate measurement precision and resolution?

The acquisition precision of the laser data is thoroughly analyzed and described in the given reference Kaasalainen et al., The Cryosphere, 5, 135–138, 2011. They state that "The standard deviation of the average of the 212 corresponding points from each profile increased linearly as a function of distance from about 0.7 mm at 3 m to about 2 mm in 11 m, which indicates that the system is capable distinguishing the surface roughness features in the (sub)millimetre scale."
In this manuscript the text in line 155 is edited to provide the information: The absolute precision of these measurements was analysed by Kaasalainen et al. (2011) to be better than 5 cm, while the relative accuracy (which is more relevant for observing the snow roughness) was found to be 0.7 mm – 2 mm for a static system, and better than 10 mm when the snowmobile was moving.

The plate method precision is analyzed in the paper Manninen et al., Journal of Glaciology, 58(211), 993–1007, 2012 and the text line 146 is edited to state: The vertical precision was about 0.1 mm and the horizontal precision 0.04 mm.

Line 158-160: Is it possible to provide a figure to show the snowmobile tracks? and one scanned profile? How was "the root mean square (rms) slopes per size of horizontal distance increment" calculated? What's the unit?
The snowmobile tracks are already published in Figs. 1 and 2 in the given reference Kaasalainen et al., The Cryosphere, 5, 135–138, 2011 and in Figs. 1 and 3 in the reference Kukko et al., Cold Regions Science and Technology, 96, 23-35, 2013. Examples of the laser scanned profiles (and plate profiles) are also shown in the reference Kukko et al., 2013. To help an interested reader to find this information, the text is accordingly slightly edited in line 154.

The rms slope calculation is now described in detail starting in line 162: The slope angles for successive points were determined for each scan of the whole data set. The slope angles were then binned according to the horizontal distance between the successive points, with a bin width of $10^{-5}$ m. Then the root-mean-square value of the slope angles was determined for each horizontal distance bin and a regression function for the dependence of slope angles on distance between successive points was derived. The slope angles are presented in degrees (Figs. 6 and 7), as they are more illustrative than radians.

Line 209: per cent -> percent
Edited as requested.

Line 212: "complemented by set or artificial light measurements", typos?
Yes, the typo is now corrected.

Equation (4): please make the equation and the description of variables consistent.
Upon the request by Dr. Picard in the short comments the number of reflections of single photon and an ensemble of photons are now explicitly denoted differently: $n$ and $m$ for single photons and $<n>$ and $<m>$ for the ensemble means. The Eqs. 3 and 4 are revised accordingly.

Line 313-314: provide the surface height distribution histograms like figure 4.
Figure 4 is revised now to include both the rms slope and the surface height distributions.

Equation (7) and (8): give a detailed description of each variable shown in the equations.
It is checked that all variables are now given, $\theta_i$ and $\theta_o$ were added.

Line 356-369: This section involves lots of statistical correlation analyses between surface roughness parameters and surface albedo measurements.  Please provide the scatter plots as supplementary material to show the correlations.  The authors used the rms height as the only explanatory variable to explain the variability of albedo and obtained quite high R square and therefore concluded that "This result supports theview that surface roughness affects the albedo."  I don't think this is a solid statement given that the snow grain size variations from March to April could greatly affect the variability of surface albedo.  Without controlling for the snow grain size, it is not plausible to validate the relationship between rms slope and surface albedo.  The authors should take both rms slope and snow grain size as the explanatory variables to explain the variability of surface albedo, and then compare the relative contributions of rms slope and snow grain size.

Scatterplots related to the text are now shown in the supplementary material document.

Definitely grain size variation from March to April affects the albedo, that is known in advance. Although both the surface roughness and grain size on the average increase with aging of the snow, the roughness does not depend only on the grain size. Otherwise there could not be any anisotropic surfaces, which typically appear because of the wind. The statistical correlation of Figure 8 is not said to be a proof of causality, it just supports the view that surface roughness affects the albedo and hence motivates closer analysis of the surface roughness and albedo relationship. The text is slightly edited in line 394: While correlation is not a proof of causality, this result supports the view that surface roughness affects the albedo.

Carrying out simple regression with both surface roughness and grain size is not worth doing, since the effect of those parameters on albedo is much more complex than a linear regression would show. (Besides, with grain size one would have to consider, whether to use only the surface grain size or also the grain size values deeper down in the snowpack.) Hence, the united analysis of grain size and surface roughness is carried out using the TARTES model (grain size) and the photon recollision model (surface roughness) as shown in section 3.

Discussion: according to the analysis, it appears that the spatial scale matters when evaluating the impact of surface roughness to snow albedo.  Based on the results, can the authors discuss the implications of this study for studying the snow albedo changes using satellite data of different resolutions? Is there a rough estimate on how much surface roughness would affect the satellite albedo (such as MODIS albedo)?
This question is really interesting, but one would need a whole new study to answer it properly.

The scale matters when estimating the numerical values for rms slope angles, which are completely scale- dependent. However, in general snow is fractal and the surface roughness can be characterized with scale-independent variables (a, b, k of Eqs. 1 and 2).

If the satellite pixel consisted of an ideal plane of, for example, 100 x 100 one square meter patches of snow with equal properties, the satellite would observe the same albedo for the whole pixel as is measured in situ for a one square meter patch. But the question is, what happens to a pixel of (fractally) undulating surface. The surface roughness effect may then be larger, when one takes into account also others roughness scales than the smallest scale.

To summarize: the principle is the same in satellite resolution, but the question is, how to estimate the multiscale average number of facet-to-facet scattering. The effect may be slightly larger in satellite resolution than at in situ measurement resolution, but probably typically not radically,

because of the fractal nature of snow. Because the rms slope angle decreases with increasing scale, it is expected that facet-to-facet scattering events across large horizontal distances are rare. Maybe a second order effect would come from including larger spatial scales?

The following text is added to the end of the discussion section:
An interesting question is how the impact of surface roughness depends on the horizontal scale, especially considering the resolution of satellite measurements. It is expected that for an ideally flat surface, the impact of roughness would be essentially the same at the satellite resolution as that in the scale of in situ measurements. However, the larger the satellite pixel is, the larger spatial scale roughness has to be taken into account. The derived model is applicable to take into account roughness of all relevant scales, but the problem is how to estimate the average multiscale number of facet-to-facet scattering events. It is anticipated that due to the fractal nature of snow the small scale estimate of the average number of facet-to-facet scattering events is a reasonable first order estimate for the corresponding multiscale value, but a related detailed analysis is beyond the scope of this study.

Figure 3. Please label the figure to indicate the meaning of each different line.
Edited as requested.

Figure 11. For the April data, it appears that the uncertainty was dramatically increased after Julian day 112, any possible reason?
The temperature of the snowpack was below zero during the first measurement days in April, numbered 40-42 from the beginning of the campaigns, (i.e. Julian days 110-112) despite of the air temperature being above zero in the Julian days 111 and 112. However, in the Julian days 113-118 the temperature was zero at every depth measured in the snowpacks. Below is a figure showing the air and snowpack temperature values measured in April 2009 (SNORTEX II). The figure below is provided in the reference Manninen and Roujean, 2014. A comment related to this phenomenonon is added in line 444.

[Figure]

Figure 13. Shrink the Y axis scale to 0.5~0.8 instead of using 0-1, otherwise, the measured albedo seems to be constant through the time.
The Y axis is now shrinked to make it clear that the measured albedo is decreasing, as pointed out by the reviewer. The stepwise decrease in the modelled values is due to the limited accuracy the grain size estimation (¼ mm).

Figure 14.  Are there three different profiles for the "rough" surface?  If so, label therough profiles differently (different parameters)?

There are three different rough snow locations, each having a few profiles. They are now made distinct, but they go quite a lot on top of each other.

Answers to short comments made by Ghislain Picard

Dr. Picard's comments are shown in black and the authors' answers are shown in blue. The line numbers refer to the revised version.

Considering recent work done on albedo over snow surface roughness by Larue etal. 2020 (https://tc.copernicus.org/articles/14/1651/2020/tc-14-1651-2020.html), I have read the paper and listed some comments, with the hope to help improving this interesting paper.
The suggested paper by Larue et al. 2020 was not available, when writing of the manuscript was started, but definitely this paper is really interesting and is now added to the list of references and assessed in the introduction and discussion. Thanks for pointing this paper out!

Detailed comments:
L34. It is not clear the diameter of what it is.
The text in line 34 was edited to make this clear: Traditionally snow grain size is characterized by its maximum diameter.

L35. "the more appropriate"→"a more appropriate"
Edited as requested.

L46. Maybe consider Larue et al. 2020 here.
We added this reference and a comment to that in line 47.

L91. "were located". Do you mean sampled ?
The text says that "half a dozen test sites were located". The authors meant to say that every day half a dozen test sites were used for diverse measurements (snowpits, surface roughness, sometimes albedo, BRF etc.). The choice of the test sites was made in advance to cover a larger area and various land cover types. The text in line 92 is slightly edited to clarify this.

L107. I recommend to use S.I units and scientific notation. kg m-3
Edited as suggested.

L140. Please add a reference for "Brownian surface"
A reference (Russ, 1994) is added in line 142.

L140. Could you test this hypothesis?
Yes, the results are shown in detail in the reference Anttila et al., 2014.

L153. I'm not sure to understand this sentence.  I understand that the accuracy of the dataset is by far inadequate for the purpose of determining surface roughness.  Is this correct?
The absolute precision of these measurements, and hence the repeatability between different mobile scanning runs was analysed by Kaasalainen et al. (2011) to be better than 5 cm. This is due to inaccuracy in the positioning solution, and it would be insufficient for detecting surface roughness in a millimeter scale. However, the relative accuracy was found to be 0.7 mm – 2 mm for a static system, and better than 10 mm when the snowmobile was moving. This means that millimeter scale roughness features can be observed from a single point cloud. The text in line 154 is edited to clarify this.

L160. I suggest to clarify the processing done, it is not reproductible as written here. Present the equations ?

Because the paper is very long the authors tried to write compactly. Obviously, in some cases too compactly. This part beginning in line 161 is now written in a more detailed way:

The profiles covered an area that was 2.4 km long and the width extended into 3.2 m at both sides of the snow mobile. The slope angles for successive points were determined for each scan of the whole data set. The slope angles were then binned according to the horizontal distance between the successive points, with a bin width of $10^{-5}$ m. Then the root-mean-square value of the slope angles was determined for each horizontal distance bin and a regression function for the dependence of slope angles on distance between successive points was derived.

L165. Please indicate accuracy of the instrument and typical range in the text.

The detection limit of the thermal-optical OCEC method is 0.2 ugC and the uncertainty of the OCEC is estimated to be +/−0.2 µgC (+/-5% relative error for higher loaded samples). The relative portion (± 5%) is composed of the instrument variation and slight variations of sample deposit in-homogeneity and sample handling (Sunset Laboratory Inc., 2018https://www.sunlab.com), as we recently discussed more in detail in Meinander et al. (2020).

References:
- Meinander O., Kontu A., Kouznetsov R., Sofiev M. Snow Samples Combined With Long-Range Transport Modeling to Reveal the Origin and Temporal Variability of Black Carbon in Seasonal Snow in Sodankylä (67°N). Front. Earth Sci. 12 June 2020, https://www.frontiersin.org/articles/10.3389/feart.2020.00153/full.
- Sunset Laboratory Inc. Organic Carbon / Elemental Carbon (OCEC) Laboratory Instrument Manual. Sunset Laboratory Inc, Forest Grove. Available online at: www.sunlab.com (accessed April 8, 2020), 2018.

The text is revised accordingly in line 172.

L187. The equations used for the correction not presented, as well as some values for the order of magnitude of this correction. This would help to understand and reproduce what is done here.

The tripod effect on the reflected radiative flux was estimated to be 5.5%. The value was derived by first photographing the downward-looking pyranometer's field of view with a camera fixed with a fisheye lens, mounted on the position where the pyranometer dome would have been on the tripod assembly. The FOV blocked by the tripod legs was calculated from the photograph. By assuming a flat reflectivity of 0.1 for the tripod material, and by using previous FIGIFIGO estimates of snow HDRF at Sodankylä to account for snow reflectivity for non-blocked FOV, the direct shadowing effect was estimated at 5.5%. This value was used to correct all reflected radiative flux data from the CM-14 albedometer. The number (5.5%, previously 3 %) was critically checked during the review process and the results (Figs.9, 10, 12 and 13) are updated accordingly, but the effect is minimal and does not affect any conclusions.

L195. There are different ways to measure albedo, your method should be more precisely described (add a scheme or picture?). From what I guess, the instrument is looking downward and you take two successive measurements, one of the snow and one if the spectralon lying on the ground ? If this is correct, then the mearsurements of the clear-sky day must be denoted "bidirectional reflectance" or "biconical reflectance",not albedo. This is critical for a precise comparison. In addition, I'd need to know 1)which size and which reflectance has the spectralon 2) if a tripod was used and shadow correction was done, and 3) how the spectralon leveling was guaranteed ? This latter point is very important for the accuracy, even an invisible tilt has a huge impact during clear-sky days.

The ASD measurements snow reflectance were indeed nadir-viewing with a 8-degree foreoptic for the year 2009 measurements considered in this paper, with the fiber's pistol grip mounted on a tripod for stability. The text around L195 clearly refers to these measurements as reflectance only,

not albedo. For further clarity, the authors amended Figure 9 caption noting these data to "Variation range (grey) of the snow reflectance spectra...". ASD reflectance data was used only for checking the grain shape to be used in the TARTES model. Note also that when the TARTES albedos were compared with the ASD reflectances, a first-order correction was made to make the reflectances better comparable with albedos (lines 393-395): The ASD reflectance spectra were scaled so that the derived broadband reflectance value matched the calibrated operationally measured broadband albedo value. The scaling factor was 0.994 for the diffuse case of March 13 and 0.937 for the clear-sky case of April 22. For March the reflectance spectrum was very close to the albedo spectrum, as the illumination was very diffuse and so the reflectance was insensitive to BRDF.

To answer the additional questions,

1) The white reference spectralon (SRM-99) was rectangular and approximately 12.5 cm x 12.5 cm in size. The spectralon was housed in a separate container with two orthogonal spirit levels. The spectralon was sanded and cleaned with pure water before the campaign to ensure a pristine surface.

2) A tripod was used. First the white reference container was placed on the snow and leveled. Narrow-view foreoptics were used to ensure that the FOV fits fully onto the spectralon. This was further visually confirmed by looking through the foreoptic before inserting the fiber optic cable. The white reference was measured and then the tripod was carefully rotated so that the foreoptic pointed into pristine snow. Tripod leg shadowing on the measured area was carefully avoided for both white reference and snow measurements.

3) The spectralon was leveled within the accuracy limits of the two orthogonally oriented spirit levels in the container, but it is difficult to provide an absolute angular measure for the leveling accuracy. Acknowledging this, the authors wish to emphasize that the results in the manuscript are not dependent on the absolute calibration accuracy of the ASD-retrieved snow reflectances.

Text starting in line 204 is edited to pass main parts of this information for the reader.

L206. Same question for the spectralon as in the previous comment. Especially if the surface is rough, it seems difficult to level the spectralon without special equipment.

FIGIFIGO measures the BRF in multiple directions (typically over 100 but depends on conditions) over the hemisphere. The spectral albedo is derived from BRF data by fitting a polynomial function and integrating over the hemisphere. The Spectralon is levelled carefully using a screw adjustable mount and a bubble level. Spectralon has dimensions of 25cm*25 cm. Nominal reflectance is 99%. Here we used more detailed value measured with Mikes-Aalto (Peltoniemi et al 2014). The text in line 217 is edited to provide this information.

L207 "real broadband albedos" I don't understand, the instrument is measuring BRF,not albedos ?

See answer to the previous comment.

L 293. "per profile". I guess that the model is 2D? The results would be very different in a 3D configuration, in particular the probability of escape from the surface. Please add some precision about the configuration and the underlying approximation if it is2D.

In the submitted manuscript the 2D/3D matter was dealt with in the results part, but it is true that it should rather be here. Hence, part of the section 4.5 was moved to section 3.3 in line 317. There we describe how 3D scattering angle estimates were derived from the 2D profiles:

Since the surface roughness profiles produce only 2D information and scattering angles differ markedly in 2D and 3D, the calculated ray tracing based 2D BRFs were converted to 3D versions assuming that each facet has besides the measured vertical angle also an azimuth angle obeying a random uniform distribution between 0 and 180°. In fact, calculations were made for the range 0 – 90°, assuming the case to be symmetrical with respect to azimuth angle, like in constructing the FIGIFIGO based BRFs. The 3D conversions make the peaks of the 2D scattering angle distributions slightly less distinct.

L296. This approximation might be good for ice or water but snow is close to a lambertian surface. This is expected to greatly affect the results.

While the mirror reflection is an approximation, it is in fact often used in geometric optics calculations of ice crystal single-scattering properties in the solar spectral region (for example: Nousiainen, T., and G. M. McFarquhar, 2004: Light Scattering by Quasi-Spherical Ice Crystals. J. Atmos. Sci., 61, 2229–2248, https://doi.org/10.1175/1520-0469(2004)061<2229:LSBQIC>2.0.CO;2; Zhang, Z., P. Yang, G. Kattawar, S. Tsay, B. Baum, Y. Hu, A. Heymsfield, and J. Reichardt, "Geometrical-optics solution to light scattering by droxtal ice crystals,"Appl. Opt. 43, 2490-2499, 2004). Basically, this approach entails the assumption that snow structures important for scattering are very large compared to the wavelength, which should be mostly reasonable. Also, this approximation neglects the impact on scattering due to snow structures smaller than the resolution of measurements (which is ~0.1 mm for plate measurements), be they "large" or "small" compared to the wavelength. This is acknowledged in the revised manuscript.

Note that mirror reflection from facets would result in a BRF dominated by specular reflection (a snow glint) only for a completely flat snow surface. When the facet orientations vary, this will act to make the resulting BRF more Lambertian, though of course not completely so. Furthemore, note that while in some cases the observed snow BRF is close to Lambertian, this is not true in general. For example, the paper by Dumont et al. (2010), in which Dr. Picard is a coauthor himself, noted the following.:
   (1) *At nadir lighting, snow reflectance is nearly Lambertian. However, the anisotropy factor is not fully circularly (varying only with $\theta v$), as it should be for perfectly horizontal sample.*
   (2) *At 30°incident angle, and 0.6 μm, R shows a forward scattering peak at ($\theta v$=30°,$\varphi$=180°). This feature is referred as darkening at grazing angles in the following since R is decreasing at limb in the forward direction. R maximum increases and shifts to larger viewing angles as wavelength increases.*
   (3) *At higher incident angle (60°), the forward scattering peak becomes sharper and stronger and is observable at both wavelengths.*
In the area the measurements of our manuscript were carried out the solar zenith angle is never close to nadir, so the last of the above mentioned three alternatives is the most common case.

Mirror reflectors are one extreme and lambertian the other extreme. Real materials are something between those two extremes. New midwinter snow is typically highly forward scattering. When the snow ages, the scattering characteristics change typically towards lambertian behaviour, but also the backward scattering increases with aging (and increasing surface roughness, see Fig. 14). In midwinter it is also typical to see individual snow stars glitter ("diamonds on snow", Figure 18 bottom). A lambertian surface would not have such behaviour. Nor would it have the striped character of Figure 18 middle or the BRF of Figure 14. In large scale it is common to have features in old snow that cause sun glints. Some photos are added below to demonstrate these unti-lambertian glittering features and also the generally uneven reflectance. At ground level these features are often difficult to observe, but from the helicopter they are distinct. In principle one could model the snow surface as a mixture of mirror and lambertian surfaces (like in T. Manninen and P. Stenberg.: "Simulation of the effect of snow covered forest floor on the total forest albedo", Agricultural and Forest Meteorology, 149:303–319,2009.) However, the ray tracing calculations of 1000 profiles with a small spatial increment are so time consuming that in practice it was possible to do only the mirror reflection alternative, which is closer to the observed snow characteristics than the other extreme. In addition, lambertian assumption for facets would have produced a lambertian BRF, which the measurements and photos do not support.

[Figure]

April 3, 2009 (T. Manninen, E. Jääskeläinen & A. Riihelä (2019): Black and White-Sky Albedo Values of Snow: In Situ Relationships for AVHRR-Based Estimation Using CLARA-A2 SAL, Canadian Journal of Remote Sensing).

A limited analysis was made about the first scattering event of the profiles. Lambertian reflection of each facet of a profile was directed to +-90 degrees of the facet normal. Out of that the amount directed in the range -90 … + 90 degrees (horizontal surface normal being 0 degrees) is the first order reflection to the sky. Below is an example of a distribution of the facet normal angles of one measured profile in March and April (the first and last measured ones of the whole set).

[Figure]

The first order scattering direction distributions for them using Lambertian facet assumption are below.

[Figure]

The fraction of lambertian multiple scattering was only 0.56% in March and 0.57% in April, hence the total BRF is dominated by the first order scattering. The facet normal angle distribution of the whole data set below is very similar to that of a single profile.

[Figure]

Hence, the conclusion is that for the studied snow a Lambertian facet assumption would result in a Lambertian total BRF, which obviously was not the case, as manifested by the photos of diverse scales. If a real surface were treated as a mixture of lambertian and mirror charateristics, the mirror reflection part would be the cause for any peaks in the BRF. Hence, analysing the BRF of purely mirror reflections shows qualitatively, how the balance between backscattering and forward scattering changes with solar zenith angle and surface roughness.

In response to this comment, the authors do not see any justification for assuming the measured snow surface to be Lambertian and they keep there original geometric optics approach. The text is slightly edited in lines 310-315.

L312. Could you define "rms slope"? Maybe for sake of clarity write "rms slope angle". However, as a general matter, I would recommend not to work with rms slope angleat all, because in theoretical studies (e.g. microwave or optical scattering by roughsurface), the variable that naturally appears in the equations is the rms gradient (in m/m not in∘), sometimes call quadratic mean slope or even rms slope in the radar altimetry community.

We use in this study the root mean square of the arcus tangent of the slope (slope = $\Delta z / \Delta x$). The arcus tangent of the slope is called the slope angle. The reason for using the slope angle rather than the slope is simply that it produced better results than the slope itself. The reason is probably, that in the high resolution roughness profiles it is possible to get perpendicular or almost perpendicular slopes (i.e. $\Delta x \approx 0$). Then the slope approaches infinity, but the slope angle remains finite. In any case

rms slope (or the rms slope angle) is not a parameter one would like to use, because it is completely scale dependent, but other roughness related parameters did not produce as good results. We checked the text that we use the term slope angle everywhere. In addition, in line 145 we added the comment that the slope angle is the arcus tangent of the slope.

L320.Is it relevant to propose empirical relationships when the literature provides analytical solutions of the albedo of Gaussian rough surfaces (e.g.doi:10.1029/2012JD018181)? The latter has also the merit to assume a lambertian surface which better applies to snow than the specular/mirror approximation.
It is relevant to propose empirical relationships, because the studied individual snow surfaces are not necessarily Gaussian and they are not typically lambertian. In this manuscript the Gaussian regression to the heights was made taking all data per month in the same distribution to get a general view of the monthly ensemble (about 3 million individual heights in March and 2.2 milloin individual heights in April). However, individual profiles (on the average 3800 individual heights) could deviate quite a lot from Gaussian height behaviour, despite of the large statistics. This was not made clear in the manuscript, and to emphasize this, statistical values for the ratio of the skewness and standard deviation per profile are now added in the text in line 336. They are also given now per standard deviation, since the ratios are more illustrative than just the skewness values.

But it is more important to notice that a Gaussian **height distribution** that is mentioned in the text (line 336) does not automatically guarantee a Gaussian **autocorrelation function (ACF),** which is the assumption of the publication of Löwe and Helbig (2012). Any permutation of the height values produces the same height distribution, but the ACF will differ. Moreover Löwe and Helbig also assume that the surface is stationary, which does not in general apply to natural surfaces. For the studied snow profiles the single scale (stationary) Gaussian ACF produced the best fit of six alternatives only in 16% of the cases, mostly in April (see Anttila et al., 2014). Hence, the results of the paper by Löwe and Helbig can't be applied to the snow surfaces studied, as the assumptions behind the formulas do mostly not apply. To make clear the distinction between the height distribution and the ACF type, a comment and reference was added starting in line 336.

L359. This sentence seems odd. If the surface is Gaussian, the ratio sigma/L is directly related to the mean quadratic gradient (equal with a factor or sqrt(2)).  The gradientbeing the tangent of the slope, one should expect a very good relationship betweenthe ratio sigma/L and the RMS of the slope angle. Why this is not the case here?
The surfaces are not all Gaussian, whether the reviewer means the height distribution or the ACF, see the above answer. Especially the ACF is mostly not Gaussian. The single scale Gaussian ACF presented the snow surface roughness best only in 16% of the cases, mostly in April (see Anttila et al., 2014).

L 366-370. "This result supports the view" is wrong. The correlation here can be explainby the fact that over the season grain size is changing in parallel with roughness, and that grain size has a large and well known impact on albedo. The statistical correlation here is inadequate to conclude on a causal relationship.
Although both the surface roughness and grain size on the average increase with aging of the snow, the roughness does not depend only on the grain size. Otherwise there could not be any anisotropic surfaces, which typically appear because of the wind. The statistical correlation of Figure 8 is not said to be a proof of causality, it just supports the view that surface roughness affects the albedo and hence motivates closer analysis of the surface roughness and albedo relationship. The text is slightly edited in line 395: While correlation is not a proof of causality, this result supports the view that surface roughness affects the albedo.

L375. I don't understand the very wide range of variation of measured albedo shown in Fig 9 given that a normalisation is done? Can you explain. Moreover, please provide more information on the normalisation

Please, notice that the individual albedo/reflectance curves were measured in individual test sites, and all curves per day are included in the grey band. The snow (surface roughness or grain size) is not identical in them. Especially in the latter figure (Fig. 9b), the variation is large as the site is a wetland and timing is that of late melting season (see Fig. 15). In April 22, the density of the topmost layer varied there in the range 0.15 – 0.43 g/cm3 and the corresponding minimum grain size varied in the range 1 – 2 mm. In March the test sites were in a forested region with clearings, where the diurnal temperature variation depends also on the canopy. The density of snow in the topmost 10 cm varied there in the range 0.11 – 0.18 g/cm3 and the corresponding minimum grain size varied between 0.5 – 1.5 mm. The normalisation coefficient values are now provided in the text in line 403. They are the ratios of the measured albedo to the average reflectance weighted with the solar irradiance. The scaling factor was 0.994 for the diffuse case of March 13 and 0.937 for the clear-sky case of April 22

L390. For clarity here add "using the TARTES model assuming a flat surface"
Edited as suggested.

L393-395. How the (huge) uncertainty on grain size measurements is taken into account here to draw this conclusion? It seems in addition that the selection of the grainshape based on the quality of the simulation/observation in Figure 9, interferes withthis conclusion.  By choosing another grain shape, a different conclusion could havebeen easily drawn, isn't it?

Obviously this text was not written clearly enough. The photos available at every measurement site suggested that spheres would be a good choice in April, but definitely not in March. Figure 9 then supports the choice: fractals for March and spheres for April. The text is slightly edited in line 415.

The uncertainty of grain size estimation is discussed in the answer to the next comment.

L411. Do you reach the same conclusion with different grain shapes? How sensitive is this to grain size uncertainty? More specifically, I'd like to see the result of March with spheres

The grain shape choice in the TARTES model calculations mattered only little for the snow studied, except that spheres produced clearly lower values than the others. This means that in April the difference would have been even larger, if some other grain shape would have been used. In March the modelled albedo value would have been smaller, if spheres had been used, but the photos taken do not justify using spheres, neither the spectra measured, see previous comment.

Below is a series of photos from the first test site of every measurement day in March 2009. The grid resolution in the photos below is 1 mm.

[Figure]

[Figure]

March 11, 2009.                                          March 12, 2009

[Figure]

March 13, 2009

March 14, 2009

March 15, 2009

March 16, 2009

March 17, 2009

March 18, 2009

March 19, 2009

On the basis of the images there is no justification for using spheres for March, hence Figure 12 will not be processed for spheres in March, but the authors made a small sensitivity analysis concerning the results of March by calculating the albedo for the first individual measurement point with the TARTES model both for fractals and spheres for the original grain size and then for grain size values,

for which 0.25 mm, 5 mm, 0.75 mm or 1 mm was added to the original value. The result is shown in the figure below. Even using spheres one would have to add more than 0.5 mm to the measured grain size value (0.75 mm at the surface and 1 mm in the top 10 cm) to reach the measured albedo value 0.8. And for fractals one would need an addition even larger than 1 mm. Although the grain size estimation is not very accurate, this large the error can't be using a grid of 1 mm remembering that the grain size dimension relevant for scattering is not the largest diameter, but the optical equivalent value.

[Figure]

In April the grains were already so rounded and big that the uncertainty of the grain size was not an issue. Several people carried out the grain size measurements and no systematic operator dependence was noticed. The same results were also obtained by a grain size expert analyzing independently the photos afterwards. However, in March the grain size/shape was a challenge, because the grain shapes were so complex that the grain size is difficult to define precisely, although still no systematic differences were noticed between different observers. Thus, in March the uncertainty of the grain size/shape may have caused some of the difference between the measured and modeled albedo level, but not but not all of that. Thus, the difference between modelled and measured albedo cannot be fully accounted for by grain size uncertainty, even if assuming spherical grains. The text is slightly edited accordingly in line 415:
The grain size estimation uncertainty was not significant in April, because the grains were already very rounded, but in March the definition and estimation of the grain size was challenging. However, even adding 1 mm to the estimated fractal grain size would not in all cases provide the measured albedo value using the TARTES model without contribution of surface roughness. As the median grain size of the top layer was in March 0.5 mm and the corresponding maximum value was 1. 5 mm (**Error! Reference source not found.**), it is in practice highly improbable to make an error of 1 mm or more using a graded plate with 1 mm scale.

L437. I don't understand what is meant by "bulk volume scattering". In the microwaverange, surface and volume scattering are clearly two distinct mechanisms that clearly appears in the equations. In the case of a diffuse medium like snow in the optical domain, surface scattering (in the microwaves meaning) is absent, only the volume does scatter the light (except if an ice lens is present at the surface). However, the layer truly contributing to the reflection is so thin (<1 cm) that calling the mechanisms "surface scattering" is acceptable (albeit with a different meaning from the microwaves). The fact that surface roughness affects the albedo, is unrelated to surface vs volume scattering. The shape of the surface (i.e. surface roughness) is important even for volume scattering. I suggest to define the concepts used in the paper, as not all the readers understand the same way these words.

The bulk volume scattering means the scattering without any surface roughness contribution, i.e. assuming that the surface of the snowpack is an ideal plane. A comment is added in line 477 to clarify this.

L443. This should be indicated in the method section, see my comment above.
This part was moved to section 3.3 as suggested.

L447. Not taking into account the atmosphere (white sky) appears to be a strong assumption for the roughness. It is possible to argue that in the blue range (where atmosphere has an effect) albedo is close to 1, so no roughness effect is expected (your equation or see Larue et al. 2020).
For albedo the roughness would not have an effect, if the smooth surface albedo were unity, since Eq.3 trivially shows that in that case (in blue range the albedo is close to this), the rough surface albedo would be 1, since $1^{n+1} == 1$ for n = 0,1,…. However, the BRF, not albedo, is discussed here, and if the atmosphere were taken into account, it would mean that the local solar zenith angle distribution would be different for a rough and a smooth surface, and consequently the BRF might differ, even if the albedo were 1. The major difference between Figs. 14 and 16 is that the volume scattering contribution is included only in the former.

L450-455. It seems that the text is comparing Figure 16 (the modeling result) withFigure 14 (the measurements), but this is not clear. It is also not clear why the x-axis and y-axis are different between the figures. Why the probability of reflection predictedby the model could not be converted into BRF? This is not explain in the method section. Ultimately if the goal is to provide a comparison, you should merge both figures.
The comparison is possible only qualitatively, because 1) the ray tracing results of Figure 16 contain only surface roughness contribution, i.e. the scattering produced by ideally smooth snowpack is missing, but FIGIFIGO measurements naturally contain contributions both from the volume and surface roughness, 2) FIGIFIGO was not measuring exactly the same snow, only in the same area and 3) FIGIFIGO sample size is 10 cm x 10 cm, whereas the surface roughness profiles were 1 m long. Therefore there is no need to make the axes equal either. The point is that the theoretical curves of Figure 16 show relatively high backscattering contribution from surface roughness that increases with the solar zenith angle. And the empirical curves of Figure 14 naturally show much milder variation as the smooth snowpack volume scattering dominates, but still they show increase of backscattering contribution from smooth to rough surface.

L475. The sentence needs to be reformulated with referring to "p", which is only defined in the Appendix.
The definition of p (photon recollision probability) is written here as requested.

L486 – 504. I don't understand this part at all. What do you mean by "it may be that a surface layer containing very deep pits would benefit from some special attention." or"the deep pit structure of the surface," ? What is "pits" here? What size / depth are youreferring to?
Here "pits" mean "cavities" (we replaced the word in the text to avoid confusion with snowpits). Deep pits are typical in forested areas, when the snow on the crown melts and drops on the forest floor. At the end of the melting season the snow always looks spongy in the vicinity of trees (see photos below), but also in midwinter after a temporary melting event that happens. In addition, in spring all dark material on the snow acts to melt surrounding snow due to light absorbtion, so that those objects (for example tree needles or litter) sink in the snowpack leaving a light trapping channel above. The pits caused by melt water drops or needles are in the mm – cm range.

[Figure]

[Figure]

April 28, 2009, forest floor

Also traces of animals (weasels, reindeer etc.) leave pits. Depending on the animal their scale is from about 1 cm upwards.

[Figure]

Mouse tracks and bird wing imprints in snow; Jane Olson; January 2016; Catalog #20490d; Original #IMG_1709. (wikimedia commons).

Section Discussion. I suggest to discuss your results w/r to results in the literature (e.g.Warren et al.1998, L'Hermitte et al., 2014 and Larue et al. 2020). The latter study has similar goal and scale to yours.

The discussion is now revised starting in line 515 to compare with the mentioned publications: The findings that surface roughness in general decreases albedo and that the effect is larger for larger solar zenith angles are quite in line with the recent results obtained by applying a new rough surface ray-tracing (RSRT) model to artificially generated surface roughness of snow (Larue et al., 2020). This study, however, extends these findings to smaller-scale roughness down to the submillimetre-scale. The discussion about the effect of the varying local incidence angle on a rough surface and shadowing effects has been going on decades, but until recently the emphasis has been on large-scale features (Warren et al., 1998; Kokhanovsky and Zege, 2004; Lhermitte et al., 2014). The essential advantage in studying the roughness from the theoretical point of view by generating artificial roughness with known dimensions and orientation is that then one can study the effect of each parameter involved separately (Larue et al., 2020). However, it is not trivial to generalize those results to natural snow, because the deterministic periodic structures may generate scattering features that will not be present for scattering from randomly rough surfaces. The advantage of the

statistical approach presented in this study is that it does not make assumptions about the surface roughness characteristics but deals with the surfaces found in nature. In addition, the derived formulas of the rough surface albedo are mathematically very simple and depend only on very few parameters, which makes their use very easy.

Although the surface scattering model used here includes multiple scattering from the surface, it may be that a surface layer containing very deep cavities would benefit from some special attention, like in the case of large-scale penitentes (Lhermitte et al., 2014).

L585. The total amounts of absorbed and scattered signal is obtained when n tend to infinity ? "n" should be replace by infinity.
The formulas for n=infinity are on page 22 (A20 and A21). Both the finite and infinite formulations are needed, when deriving Eq. A24. Hence, the authors keep the presentation of the approach as is, but edit the phrasing to discriminate between finite and infinite cases of n.

L591. "n" was not defined (it was defined L261 in a different way), so it is difficult to understand n=0. Ps=0 si clear and it implies than any terms for n > 1 has zero contribution. But n=0 is not clear (see next comment).
The symbol n is defined in the previous line 646 in the Appendix: … "the number of additional facet scattering sequences (n) the photon has before it escapes altogether…" Then n=0 means that no additional facet scattering takes place, i.e. it corresponds to the ideal case of completely smooth snowpack with only volume scattering. See the next answer.

L602. "the average value for n" contradicts the definition given L216 where "n" is the average of the number of event. I suspect that a different symbol should be used between the number of event before escaping and the average of this number. (A25) uses "n" which is in the number of terms in (A17 and A18) for a given photon. It can not be "changed" to the average number of "n" for all the photons. This is equivalent to switch sum and product...and results in the strange result (A27) which I don'tunderstand (intuitively). I don't know if/where is the error but using "n" in two differentways seems risky.
The notation is already rather heavy, but it is important to make things clear for the reader. Hence, the average value of n is now denoted explicitly by <n>. Precisely theoretically the distribution of n should be used to derive the average albedo (as is now described in the added text starting in line 657) instead of just using the mean value in Eq.3, but in practice the observed empirical distributions indicate that the effect of using the distribution of n instead of just <n> is very small. The empirical distributions for all data of March and April are shown in the figures below and corresponding albedo estimates based on those distribution and <n> are compared in the two following figures. Since the smooth surface albedo of snow is high (larger than 0.8 in March and larger than 0.7 in April), the use of <n> in albedo estimation is justified. Determining reliably the distribution of n for single profiles would be very challenging, because the values n=0 and n=1 dominate. Probably the albedo estimation would be more inaccurate due to that than when using values of <n> for single profiles.

[Figure]

The (admittedly, surprisingly simple!) equation A29 follows from rigorous theoretical formulation shown in detail in the Appendix. The key to understanding the derivation is the text between Eqs. (A24) and (A25). In the revised version of the paper, an effort was made to make this text more explicit and clearer. We agree it is of paramount importance to check the equations properly. For that reason several of the co-authors checked them independently and also the 1st reviewer considers the approach sound and the 2nd reviewer did not critisize it. Therefore, unless the commenter is able to show that something is actually wrong (saying that something is non-intuitive falls short of that), we will stick to our position that the equations are correct.